# Coupling thermotolerance and high production of recombinant protein by CYR1^N1546K mutation via cAMP signaling cascades
Haiyan Ren[1,2], Qing Lan[1,2], Shihao Zhou[1,2], Yilin Lyu[1,2], Yao Yu[1,2], Jungang Zhou[1,2], Wenjuan Mo [1,2] ✉ & Hong Lu [1,2] ✉

In recombinant protein-producing yeast strains, cells experience high production-related stresses similar to high temperatures. It is possible to increase recombinant protein production by enhancing thermotolerance, but few studies have focused on this topic. Here we aim to identify cellular regulators that can simultaneously activate thermotolerance and high yield of recombinant protein. Through screening at 46 °C, a heat-resistant *Kluyveromyces marxianus* (*K. marxianus*) strain FDHY23 is isolated. It also exhibits enhanced recombinant protein productivity at both 30 °C and high temperatures. The CYR1^N1546K mutation is identified as responsible for FDHY23's improved phenotype, characterized by weakened adenylate cyclase activity and reduced cAMP production. Introducing this mutation into the wild-type strain greatly enhances both thermotolerance and recombinant protein yields. RNA-seq analysis reveals that under high temperature and recombinant protein production conditions, CYR1 mutation-induced reduction in cAMP levels can stimulate cells to improve its energy supply system and optimize material synthesis, meanwhile enhance stress resistance, based on the altered cAMP signaling cascades. Our study provides CYR1 mutation as a novel target to overcome the bottleneck in achieving high production of recombinant proteins under high temperature conditions, and also offers a convenient approach for high-throughput screening of recombinant proteins with high yields.

Efficient expression of recombinant proteins is the ultimate goal in industrial biopharmaceutical production. This can be achieved through various methods, including optimizing expression elements (e.g., codon usage, gene copy number, promoters)[1–3], rationally engineering host cells (e.g., over-expression of chaperones)[4], and using random or semi-random mutagenesis to increase genetic diversity[5]. Random mutagenesis enables the discovery of new engineering possibilities, with effective screening means being the key to its success. Current high-throughput screening methods primarily use fluorescence intensity or enzyme activity, limiting the range of testable proteins[6]. Improving the screening efficiency of proteins that are inconvenient to assay, like nanoparticles, hemoglobin, and cytokines, remains a constant challenge.

High-level expression of recombinant proteins is stressful for host cells, causing protein misfolding, aggregation, and reactive oxygen species (ROS)

accumulation[7]. The intensive influx of newly synthesized peptides also stresses the endoplasmic reticulum (ER) system and Golgi vesicular transport[8]. High temperature induces protein misfolding, aggregation, ROS generation, and ER overload, ultimately leading to cell death[9–11]. Hence, the pressure from recombinant protein production bears resemblance to high temperature stress. Cell's stress tolerance greatly affects its capacity for recombinant protein production[4]. Certain extreme environmental conditions, including low temperature, hypoxia, microgravity, and high osmolality, can stimulate stress-responsive gene expression and physiological changes, resulting in an established new equilibrium and increased recombinant protein production[12]. However, enhancing thermotolerance to increase heterologous protein yield has been scarcely reported. It was known that overexpressing heat shock response genes in *Saccharomyces cerevisiae* (*S. cerevisiae*), without altering the environmental temperature, can increase

[1]State Key Laboratory of Genetic Engineering, School of Life Sciences, Fudan University, Shanghai 200438, China. [2]Shanghai Engineering Research Center of Industrial Microorganisms, Shanghai 200438, China. ✉e-mail: wenjuanmo@fudan.edu.cn; honglv@fudan.edu.cn

heterologous protein production by enhanced protein folding[13]. Therefore, it is feasible to adopt high temperature as a screening pressure to select strains with improved high-temperature tolerance and increased recombinant protein production.

*K. marxianus*, as a non-conventional yeast, has been highly promising for industrial applications and fundamental research due to its food-grade safety, high temperature resistance, and ability to produce high yields of recombinant proteins[14–16]. In this study, we aimed to improve both thermotolerance and heterologous protein production of *K. marxianus*. Through screening at 46 °C, a high-temperature tolerant strain, FDHY23, was successfully obtained. It also exhibited significantly enhanced yields of six recombinant proteins at 30 °C, with this advantage maintained even at high temperatures and in industrial high-density fermentation settings. Whole-genome sequencing identified CYR1^N1546K as the key mutation responsible for the improved performance of FDHY23. Introduction of this mutation into the wild-type strain also resulted in increased heat resistance and recombinant protein production. The CYR1^N1546K mutation reduces adenylate cyclase activity and cAMP generation. The underlying mechanisms connecting CYR1 mutation to the combined phenotypes of thermotolerance and high yields of recombinant proteins were essentially unveiled by the transcriptome analysis during high temperature and recombinant protein production conditions. Therefore, we propose the CYR1 mutation as a key switch in coupling high temperature tolerance and high productivity of recombinant proteins. And we also demonstrate the practicability of using high temperature as an efficient means for high-throughput screening of highly produced recombinant proteins.

## Results

### The thermotolerant strain FDHY23 is obtained by screening under high temperature condition

In this study, a thermotolerant *K. marxianus* strain, FDHY23, was obtained by subjecting the wild-type strain LHP1044 to high temperature selection at 46 °C. FDHY23 exhibited remarkably improved heat resistance, as evidenced by its higher cell viability in spot assays compared to the reference strain LHP1044 at temperatures of 46 °C, 47 °C, and 48 °C (Fig. 1a). Growth curves for strains FDHY23 and LHP1044 were analyzed at different temperatures (Fig. 1b). There was no significant difference in the maximum $OD_{600}$ reached during the stationary phase between these two strains at both 30 °C and high temperatures (Fig. 1b). However, under high temperature conditions, FDHY23 exhibited a shorter lag phase (Fig. 1b) and a significantly faster maximum specific growth rate during the logarithmic phase (Fig. 1c)

compared to LHP1044. These results demonstrate that FDHY23 is better adapted to high temperature environments and exhibits a faster growth.

### CYR1^N1546K mutation is the key determinant of heat resistance phenotype

We conducted whole-genome DNA sequencing on FDHY23 and LHP1044 strains and identified specific genetic variations in FDHY23 compared to LHP1044. The mutations on the nuclear chromosomes and mitochondria are documented in Table S1 and Table S2 of Supplementary Data 1, respectively. There were only two SNP mutations in the intergenic region on the mitochondria, while multiple mutations were identified in the coding regions of the nuclear chromosomes. The identified coding region variations were validated using PCR, confirming two genes with non-synonymous mutations (Fig. 2a). One is the leukotriene A4 hydrolase gene *LAP2*, with a G to C mutation at position 1054, resulting in the substitution of glycine (G) with arginine (R) at position 352. The other is the adenylyl cyclase gene *CYR1*, with a C to G mutation at position 4638, leading to an amino acid change from asparagine (N) to lysine (K) at position 1546.

To assess the effect of these two mutations on the high-temperature tolerance phenotype, we introduced the respective mutations into strain LHP1044 using CRISPR/Cas9 technology. The LHP1044-LAP2^G352R strain carrying the introduced *LAP2* point mutation did not exhibit any obvious alteration in heat resistance (Fig. 2b). However, the LHP1044-CYR1^N1546K strain carrying the introduced *CYR1* point mutation displayed a marked enhancement in high-temperature tolerance, with no discernible difference in growth compared to LHP1044 at 30 °C (Fig. 2b). Furthermore, when the *CYR1* gene in FDHY23 was subjected to a reversion mutation, the resulting FDHY23-CYR1 strain exhibited a noticeable decrease in growth capability at 46 °C (Fig. 2b). These results demonstrate that the *CYR1* mutation is essential for the thermotolerance ability of FDHY23.

### FDHY23 improved recombinant protein production at 30 °C

We further evaluated FDHY23's capability of recombinant protein production, using recombinant soybean hemoglobin LBA as an example. The intracellular expression levels of LBA protein were compared between FDHY23 and LHP1044 in shake flasks at 30 °C. Remarkably, FDHY23 exhibited a 2.6-fold increase in LBA production compared to LHP1044 (Fig. 3a). We compared the production levels of other five recombinant proteins in FDHY23 and LHP1044 strains in shake flasks. These include the intracellularly expressed Fowlpox virus coat protein HVP2 and Porcine circovirus coat protein PCV2, as well as the secreted Est1E esterase from yak

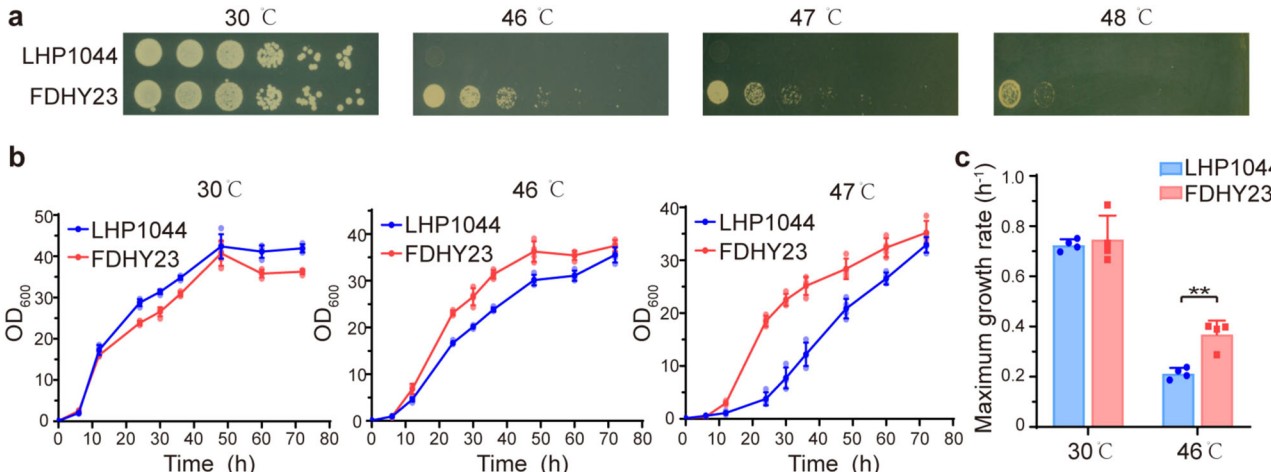

**Fig. 1 | Comparison of high temperature tolerance between strains FDHY23 and LHP1044. a** Thermotolerance spot assays. The cultures were diluted 5 times with a 5-fold dilution factor from an initial OD600 of 0.6. They were then incubated at temperatures of 30 °C, 46 °C, 47 °C, and 48 °C, respectively. **b** Growth curves. It was plotted based on the correlation between cell density ($OD_{600}$) and culture time at 30 °C, 46 °C, and 47 °C, respectively. **c** Maximum specific growth rate during the logarithmic phase. For growth curves and maximum specific growth, values were calculated as average±SD from four biological replicates. The red and blue colors represent strains FDHY23 and LHP1044, respectively. **: $p < 0.01$.

**Fig. 2 | The non-synonymous mutations detected in FDHY23 and their impact on the thermotolerance phenotype. a** Mutation positions in the *LAP2* and *CYR1* genes. The yellow marks represent the mutation sites. **b** The distinct effects of LAP2$^{G352R}$ and CYR1$^{N1546K}$ mutations on the thermotolerance phenotype. Spot assays to assess heat resistance were performed at 30 °C and 46 °C, respectively.

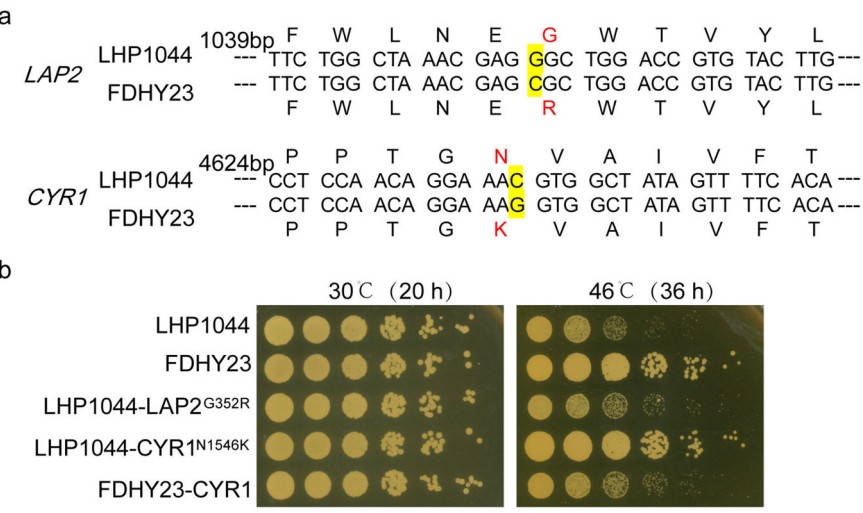

rumen, AnFaeA esterase from *Aspergillus niger*, and Badgla glycosidase from *Talaromyces purpureogenus*. As shown in Fig. 3b–f, the expression levels of HVP2, PCV2, Est1E, AnFaeA, and Badgla were 1.7-fold, 1.6-fold, 2.1-fold, 1.7-fold, and 5.5-fold higher in FDHY23 compared to LHP1044, respectively. Additionally, the intracellular retention rate of the Badgla protein was remarkably reduced in FDHY23, decreasing from 32.9% in LHP1044 to 8.9% in FDHY23. These findings demonstrate that FDHY23 is capable of producing higher levels of recombinant proteins and greatly improve the secretion process (Fig. 3g). The uncropped blot images for Fig. 3 can be found in Supplementary Fig. 1.

To determine whether the genetic mutations contribute to the enhanced recombinant protein yield in FDHY23, we expressed the recombinant LBA protein in the LHP1044-CYR1$^{N1546K}$ strain. It showed remarkably higher expression levels than LHP1044 and comparable yield to FDHY23 (Fig. 3h). The LHP1044-LAP2$^{G352R}$ strain carrying the introduced *LAP2* point mutation did not exhibit any obvious alteration in LBA production compared to LHP1044 (Fig. 3h). Therefore, the CYR1$^{N1546K}$ mutation not only enhances thermotolerance but also essentially improves recombinant protein yield.

We further analyzed the temporal production of LBA in FDHY23 and LHP1044 during shake flask cultivation at 30 °C. There was no significant difference in the growth curves of the two strains (Fig. 4a). Both strains showed an increase in LBA production over time, with a noticeable surge at the beginning of the stationary phase (Fig. 4b). Notably, FDHY23 consistently displayed remarkably higher LBA production compared to LHP1044, corroborating the above findings (Fig. 3a).

To simulate high-density fermentation in industrial settings, we investigated the expression capacity of recombinant LBA protein of FDHY23 compared to LHP1044, using a 5 L fermenter and an inorganic salt synthetic medium at 32 °C. No significant difference was observed in the growth curves between the two strains (Fig. 4c). It is worth noting that in the fermenter, both strains exhibited visible LBA bands as early as 24 h, with FDHY23 and LHP1044 reaching peak LBA concentrations of 4.79 g/L and 2.87 g/L, respectively (Fig. 4d). Therefore, the CYR1$^{N1546K}$ mutation has demonstrated the ability to effectively enhance recombinant protein production in an industrial high-density inorganic salt fermentation environment, indicating promising prospects for industrial applications. The corresponding SDS-PAGE analysis for absolute quantification is presented in Supplementary Fig. 2. The uncropped blot images for Fig. 4 can be found in Supplementary Fig. 3.

## FDHY23 demonstrates high-level recombinant protein production even at high temperatures

We further compared the ability of strains to express recombinant LBA protein under extreme high-temperature conditions (46 °C). It was observed that at 46 °C, the expression of recombinant LBA protein was barely detectable in LHP1044 (Fig. 5a), while evident expression levels were still observed in FDHY23, peaking at 60 h (Fig. 5a). Their growth curves at 46 °C showed no significant difference (Fig. 5b). The LHP1044-CYR1$^{N1546K}$ strain also expressed remarkably higher LBA production at 46 °C (Fig. 5c). These results confirm that the CYR1 mutation indeed possesses the combined phenotype of thermotolerance and high productivity of recombinant proteins. The uncropped blot images for Fig. 5 are presented in Supplementary Fig. 4.

## CYR1$^{N1546K}$ mutation weakens the enzymatic activity of adenylate cyclase and reduces intracellular cAMP levels

Through conservation domain analysis provided by NCBI, the CYR1$^{N1546K}$ mutation was identified to be located within the catalytic domain (1537–1705) of adenylate cyclase (Fig. 6a). This domain is essential for catalyzing the conversion of ATP to cAMP. The asparagine (N) at this position is conserved between *K. marxianus* and *S. cerevisiae*. When N is mutated to K, the amino acid changes from acidic to basic, which could potentially disrupt interactions with surrounding amino acids and thereby affect the catalytic efficiency of the enzyme.

We measured the enzymatic activity of adenylyl cyclase and the levels of cAMP in mid-log phase and stationary phase cells at 30 °C. Compared to LHP1044 (Fig. 6b), the adenylyl cyclase activity decreased by 36.3% and 30.6% in the mid-log phase and stationary phase cells of strain LHP1044-CYR1$^{N1546K}$, respectively. Similarly, FDHY23 showed a decrease of 34.4% and 37.2% in the same phases, respectively. The intracellular cAMP levels also showed a decrease, with strains LHP1044-CYR1$^{N1546K}$ and FDHY23 experiencing a significant drop compared to LHP1044 in the mid-log and stationary phases (Fig. 6c), respectively. Therefore, the CYR1$^{N1546K}$ mutation weakens the enzymatic activity of adenylyl cyclase and leads to a decrease in intracellular cAMP levels at 30 °C.

We further investigated the intracellular cAMP levels of strains FDHY23 and LHP1044 under two different stress conditions: high temperature (46 °C) and production of recombinant protein LBA. In the mid-log phase at 46 °C (Fig. 6d), the cAMP level of LHP1044 was significantly lower than its cAMP level at 30 °C (p = 0.009, Cohen's d value = −5.34). In contrast, the cAMP level of FDHY23 at 46 °C was higher than its cAMP level at 30 °C (p = 0.213, Cohen's d value = 1.67), with average values of 4.50 and 3.83, respectively. Since the Cohen's d value is larger than 0.8, indicating a substantial difference between the two groups, it can be concluded that the higher cAMP level in the mid-log phase of FDHY23 at 46 °C compared to 30 °C was meaningful, rather than merely occurring by chance, despite the p-value not being significant. While in the stationary phase at 46 °C (Fig. 6d), the cAMP levels of LHP1044 and FDHY23 were both significantly lower than their levels at 30 °C (p = 0.012 and p = 0.003, respectively). These

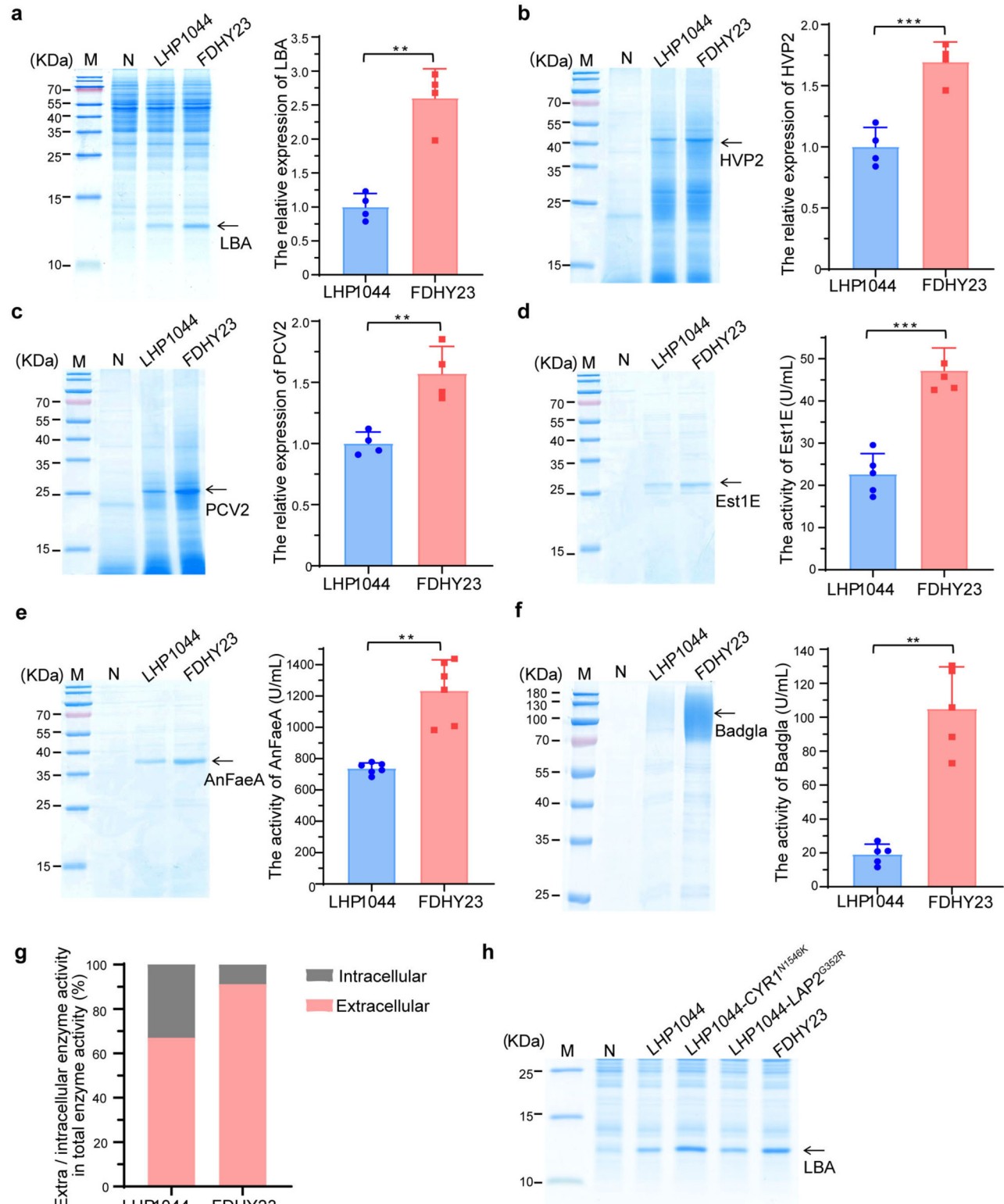

**Fig. 3 | The CYR1$^{N1546K}$ mutation improves recombinant protein yield in cells.** Strains FDHY23 and LHP1044 exhibited distinct expression levels of six recombinant proteins on SDS-PAGE at 30 °C (A-F). **a** Soybean hemoglobin LBA. Four biological replicates. **b** Fowlpox virus coat protein HVP2. Four biological replicates. **c** Porcine circovirus coat protein PCV2. Four biological replicates. **d** Est1E esterase.

Five biological replicates. **e** AnFaeA esterase. Six biological replicates. **f** Badgla glycosidase. Five biological replicates. **g** Intracellular and extracellular enzymatic activity of the glycosidase Badgla. Five biological replicates. **h** The essential effect of CYR1$^{N1546K}$ mutation on recombinant LBA protein expression. Values were calculated as average±SD from biological replicates. **: $p < 0.01$; ***: $p < 0.001$.

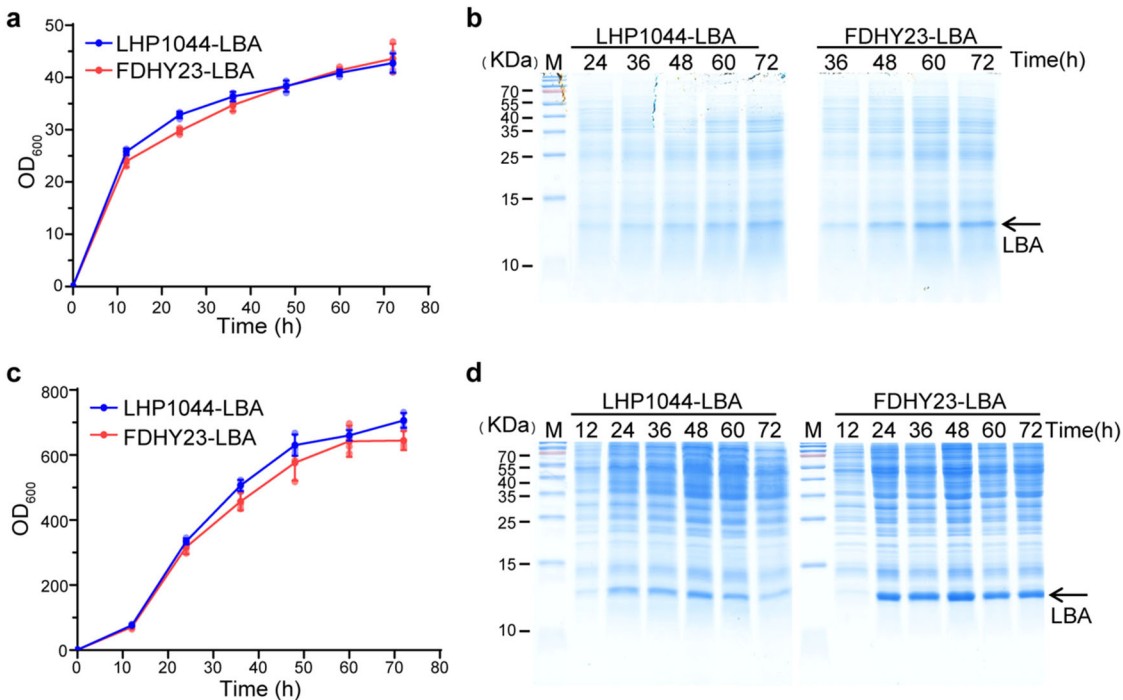

**Fig. 4 | Comparison of recombinant LBA protein temporal production between FDHY23 and LHP1044 under shake flask cultivation and industrial fermentation conditions. a** Growth curves of strains FDHY23 and LHP1044 during recombinant LBA production under shake flask cultivation at 30 °C. **b** Expressions of recombinant LBA protein under shake flask cultivation at 30 °C. LBA production was analyzed at 24 h, 36 h, 48 h, 60 h, and 72 h by SDS-PAGE. **c** Growth curves during high-density fermentation process at 32 °C. **d** Recombinant LBA protein yield during high-density fermentation with inorganic salts. LBA production was analyzed at 12 h, 24 h, 36 h, 48 h, 60 h, and 72 h by SDS-PAGE. For growth curves, values were calculated as average±SD from three biological replicates. The red and blue colors represent strains FDHY23 and LHP1044, respectively.

**Fig. 5 | Comparison of recombinant LBA protein production between FDHY23 and LHP1044 under high-temperature conditions. a** Expressions of recombinant LBA protein at 46 °C. Protein production was analyzed at 24 h, 36 h, 48 h, and 60 h by SDS-PAGE. **b** Growth curves during recombinant LBA production at 46 °C. The red and blue lines represent strains FDHY23 and LHP1044, respectively. **c** Recombinant protein yield elevation at 46 °C for the effect of $CYR1^{N1546K}$ mutation. The strain LHP1044-$CYR1^{N1546K}$, which was constructed by introducing the CYR1 mutation into strain LHP1044, also showed higher LBA production at 46 °C compared to LHP1044. For growth curves, values were calculated as average±SD from three biological replicates.

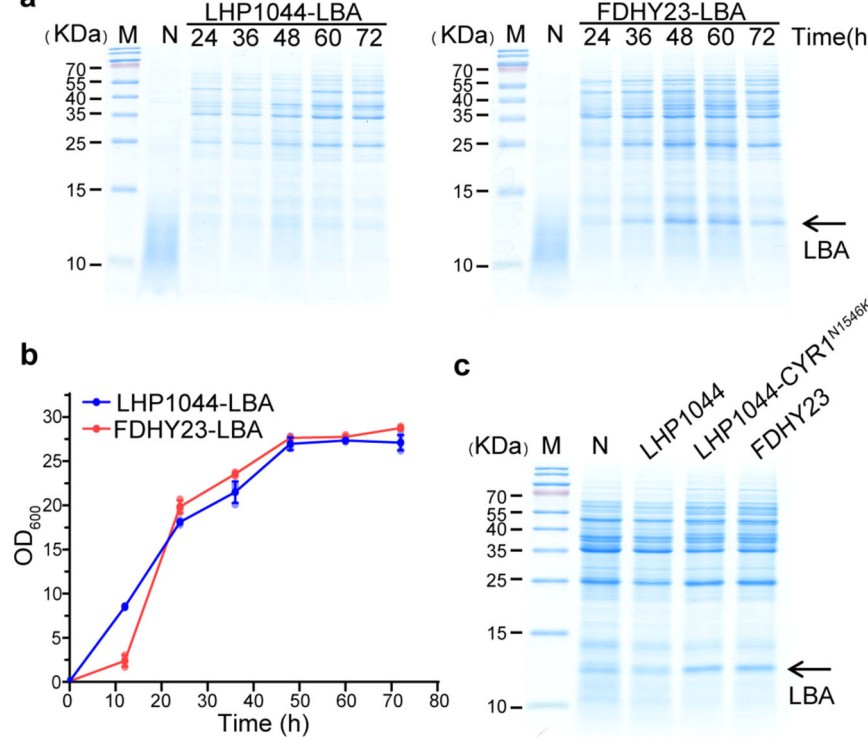

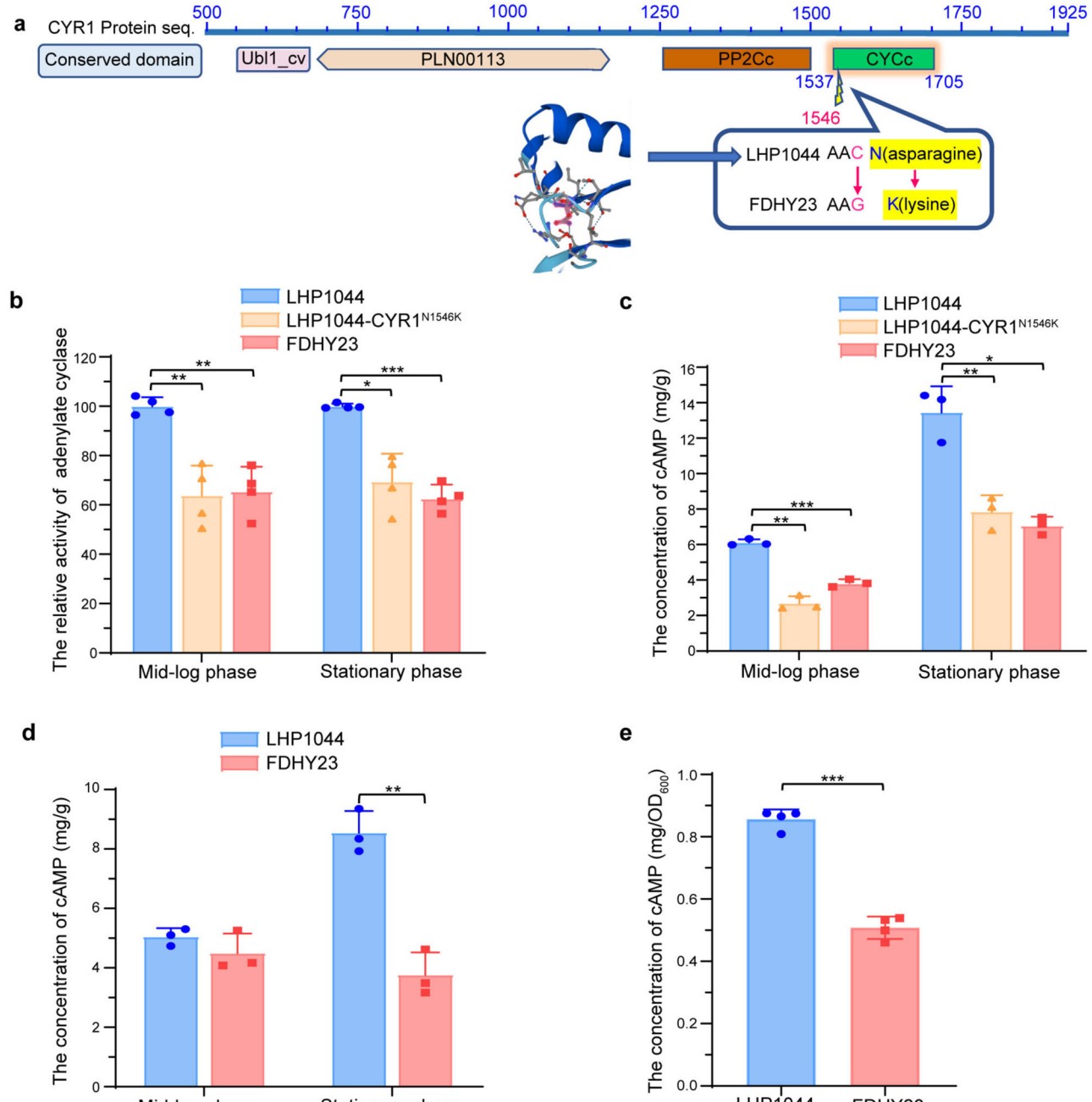

**Fig. 6 | The CYR1$^{N1546K}$ mutation affects adenylate cyclase activity and intracellular cAMP levels. a** Conserved domain containing the mutation position. It depicts the arrangement of conserved domains along the amino acid sequence of *K.marxianus*' *CYR1* gene. The mutation site at position 1546 is situated within the catalytic domain (CYCc) of the *CYR1* gene. **b** The enzymatic activity of adenylate cyclase measured at 30 °C during the mid-log phase and stationary phase of strains FDHY23, LHP1044-CYR1$^{N1546K}$, and LHP1044. Four biological replicates. **c** cAMP levels measured at 30 °C during mid-log phase and stationary phase of these three strains. Three biological replicates. **d** cAMP levels measured at 46 °C during the mid-log and stationary phases of strains FDHY23 and LHP1044. Three biological replicates. **e** cAMP levels measured at 30 °C during the stationary phase of strains FDHY23 and LHP1044 producing recombinant protein LBA. Four biological replicates. Values were calculated as average±SD from biological replicates. *: $p < 0.05$; **: $p < 0.01$; ***: $p < 0.001$.

results suggest that with the increase in temperature, FDHY23 may enhance the production of cAMP during mid-log phase, while LHP1044 significantly drops its cAMP production. For LBA production at the stationary phase at 30 °C (Fig. 6e), the cAMP level of FDHY23 remained significantly lower than LHP1044. Please note that the units for Fig. 6e differ from those in 6c and 6d. This is because when producing the recombinant protein LBA, the cell protein content cannot be used as the denominator. Therefore, the cAMP levels during LBA production (Fig. 6e) are not compared to those during normal growth (Fig. 6c).

## FDHY23 commonly enhances mitochondrial function, respiratory chain, protein folding, antioxidant response, and vesicle transport processes under high temperature and recombinant protein production conditions

To investigate the possible molecular mechanisms underlying the enhanced high-temperature tolerance and recombinant protein production capacity caused by the CYR1$^{N1546K}$ mutation, we conducted transcriptomic RNA-seq analysis on strains FDHY23 and LHP1044 during two conditions: during the mid-log phase at 46 °C

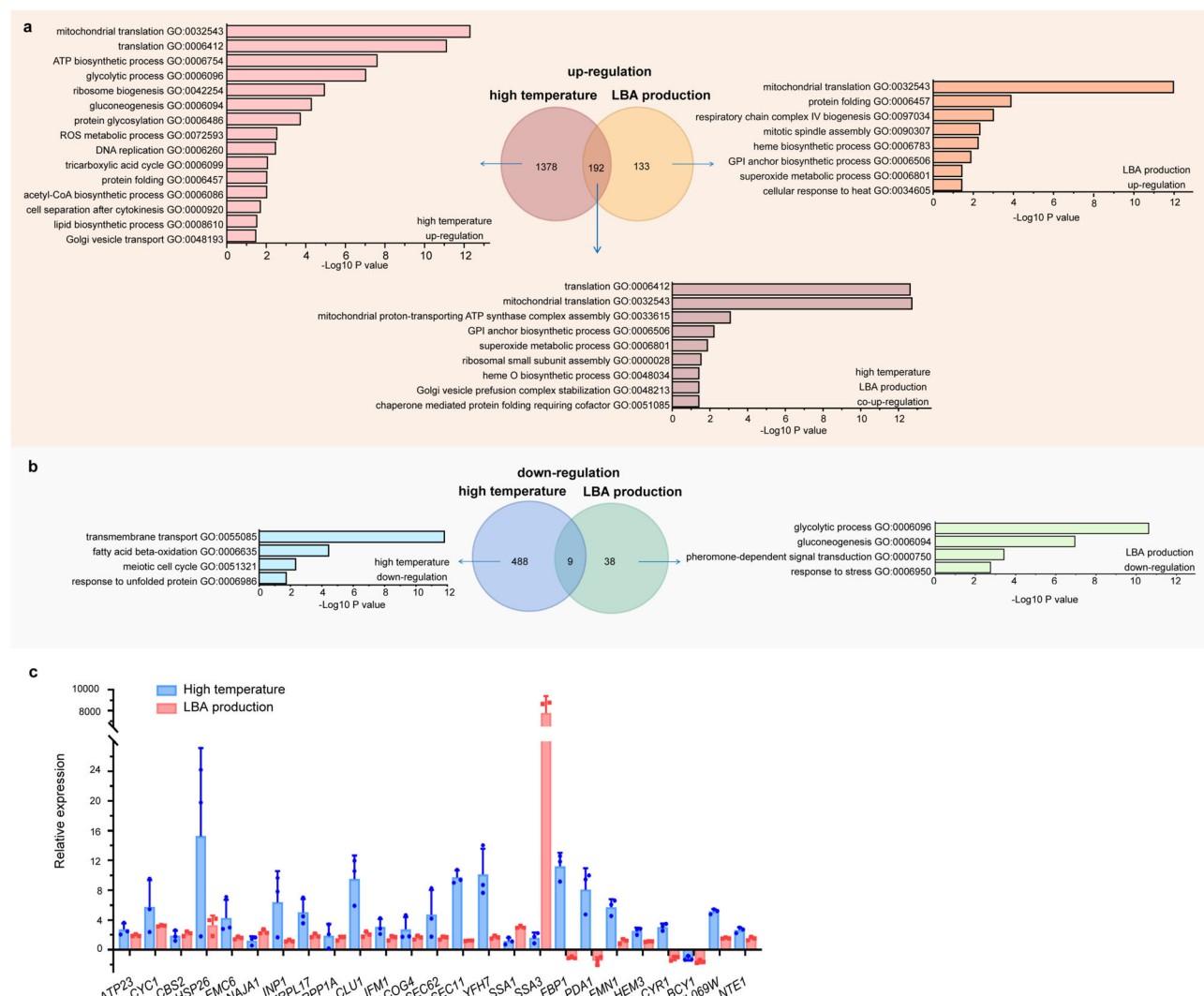

**Fig. 7 | GO enrichment analysis of differentially expressed genes between FDHY23 and LHP1044 under high temperature and recombinant protein production conditions. a** GO enrichments of upregulated genes in FDHY23 compared to LHP1044. 'LBA production' represents the recombinant protein production condition. The enrichment results were denoted as follows: pink for all genes upregulated at high temperature, orange for all genes upregulated during LBA production, and coral for genes commonly upregulated under both high temperature and LBA production conditions. **b** GO enrichments of downregulated genes in FDHY23 compared to LHP1044. Light blue and light green represent the downregulated genes at high temperature and LBA production, respectively. The numbers of differentially expressed genes are labeled in the Venn diagram. **c** qRT-PCR analysis of 25 crucial genes involved in various cellular processes. The red and purple colors represent the high temperature and LBA production conditions, respectively. Values were calculated as average±SD from three biological replicates.

(Supplementary Data 2) and during the stationary phase while expressing recombinant protein LBA at 30 °C (Supplementary Data 3). The mitochondrial-encoded genes showed no significant expression changes, therefore the subsequent analysis primarily centered on nuclear-encoded genes.

At 46 °C, FDHY23 exhibited upregulation of 1570 genes (Fig. 7a) and downregulation of 497 genes (Fig. 7b) compared to LHP1044. The upregulated genes were enriched in GO pathways related to mitochondrial respiratory chain ATP generation, ribosome biogenesis, protein translation, protein folding, glycolysis, gluconeogenesis, TCA cycle, acetyl-CoA generation, lipid synthesis, vesicle transport, protein glycosylation, ROS metabolism, DNA replication, and cytokinesis. The downregulated genes were mainly associated with transmembrane transport, fatty acid beta-oxidation, meiosis, and unfolded protein response. During recombinant protein production, FDHY23 showed upregulated 325 genes (Fig. 7a) and 47 downregulated genes (Fig. 7b) compared to LHP1044. The upregulated genes were enriched in GO pathways such as mitochondrial translation, respiratory chain assembly, heme biosynthesis, protein folding, cellular response to heat, GPI anchor biosynthesis, spindle assembly, and peroxide metabolism. While the downregulated genes were enriched in processes related to glycolysis, glycogen biosynthesis, hormone-mediated signaling, and stress response. There were 192 genes commonly upregulated in high temperature and recombinant protein production conditions, enriched in GO pathways such as protein translation, protein folding, mitochondrial function, respiratory chain ATP synthase assembly, heme biosynthesis, peroxide metabolism, Golgi vesicle fusion, and GPI anchor biosynthesis. Only 9 genes were both downregulated, mainly involved in lipid degradation and vacuole function. Interestingly, in both high temperature and recombinant protein production conditions, there were remarkably more upregulated genes than downregulated genes, indicating that the upregulated cellular processes (Supplementary Data 4) are crucial in facilitating these favorable conditions. Therefore, the commonly upregulated processes of mitochondrial function, respiratory chain ATP generation, protein folding, ribosome biogenesis, ROS resistance, and vesicle transport likely predominantly contribute to the coupling of high temperature tolerance and high recombinant protein production phenotypes.

It should be noted that during recombinant protein production, there were fewer differentially expressed genes compared to high temperature conditions. This is likely because the transcriptome during LBA production captures the stationary phase rather than the logarithmic growth phase observed at 46 °C. As a result, in the stationary phase of recombinant protein production, while metabolic changes observed at high temperature may not be sustained, key functions involved in cellular apparatus such as mitochondrial assembly, respiratory chain, protein translation and folding, ER and Golgi system are retained.

We further conducted quantitative real-time PCR (qRT-PCR) analysis of 25 crucial genes (Fig. 7c) involved in various cellular processes, including the respiratory chain, ATP generation, protein folding, anti-ROS, ribosome assembly, protein translation, secretory pathway, vesicle transport, central carbon metabolism, and cAMP signaling cascade. The qRT-PCR results were mostly consistent with the RNA-seq data (Supplementary Data 2 and Data 3), thus confirming the reliability of our RNA-seq findings.

### At high temperatures, FDHY23 reconfigures carbon flux to enhance the production of ATP, glycogen, acetyl-CoA, lysine, and nucleotides

As mentioned earlier, in strain FDHY23 compared to LHP1044, during the mid-log phase at high temperatures, there is an upregulation of genes involved in both glycolysis and gluconeogenesis (Fig. 7a), indicating the activation of two opposing cellular processes. This raises curiosity about how FDHY23 efficiently allocates carbon sources during rapid growth under high-temperature conditions. To address this, we conducted a systematic analysis of differentially expressed genes related to metabolic pathways at 46 °C (Fig. 8).

As shown in the top portion of Fig. 8, glucose uptake genes (HXT4, STL1, and SKS1) and glucose phosphorylation genes (GLK1 and RAG5) were significantly upregulated, resulting in increased production of glucose-6-phosphate. For glucose-6-P, it can be directed towards four downstream pathways, with glycolysis being the primary pathway leading to pyruvate production. Several steps in glycolysis are also shared with gluconeogenesis. Shared genes involved in both glycolysis and gluconeogenesis, such as FBA1 and PGK1, were significantly upregulated. Additionally, specific genes related to glycolysis (e.g., PFK1, PFK2, and PYK1) and gluconeogenesis (e.g., PYC2, PCK1, and FBP1) were also upregulated, indicating an overall enhancement of both glycolysis and gluconeogenesis processes.

Acetyl-CoA production from pyruvate was upregulated in both the mitochondria and cytoplasm pathways at high temperatures. In the mitochondrial pathway, the conversion of pyruvate to acetyl-CoA through the upregulation of the pyruvate dehydrogenase complex (PDA1, PDB1, and LAT1) was observed. Furthermore, there was an upregulation in the synthesis of the necessary coenzyme lipoamide. In the cytoplasm pathway, the conversion of pyruvate to acetaldehyde, and then to acetate and finally to acetyl-CoA, was upregulated at each step. While the degradation of acetyl-CoA to acetate was downregulated. It is worth noting that the enhanced conversion of acetaldehyde to acetate was accompanied by the generation of NADPH from NADP+, providing sufficient reducing power for many biosynthetic reactions, including lipid and DNA synthesis. Additionally, the conversion of acetaldehyde to ethanol in the cytoplasm (coupled with converting NADH to NAD+) and the generation of ethanol from acetaldehyde in the mitochondria (accompanied with NAD+ to NADH) were both upregulated. This "acetaldehyde-ethanol shuttle" provides NAD+ for the cytoplasmic oxidative breakdown processes like glycolysis, while also supplying sufficient NADH as an electron donor for the mitochondrial respiratory chain.

Acetyl-CoA can be used for lipid/sterol synthesis or enter the tricarboxylic acid (TCA) cycle (Fig. 8). We observed upregulation of genes involved in both processes (Supplementary Data 4), indicating increased acetyl-CoA production by FDHY23 at high temperatures. The increased synthesis of lipids and sterols can enhance membrane stability. The upregulation of the TCA cycle provides electrons for the respiratory chain and building blocks for biosynthesis. Additionally, lysine and heme synthesis, branching off from the TCA cycle, were also upregulated. Heme promotes the respiratory chain, while lysine facilitates the generation of acetyl-CoA from pyruvate, lipid synthesis, and protein translation activation.

For glucose-6-P, its second fate is to continue gluconeogenesis and produce four different forms of glycogen chains, including trehalose, branched glycogen, cell wall insoluble β-glucan, and N-glycan on glycoproteins (shown in the upper left part of Fig. 8). All four downstream pathways were upregulated, while glycogen degradation into glucose was downregulated. Intracellular branched glycogen and trehalose serve as thermoprotectants, stabilizing the cell membrane and preserving the conformation of intracellular proteins[17]. Genes BGL2 and EXG1 were upregulated, facilitating the hydrolysis of linear β-1,3-glucan on the cell wall. Conversely, the gene YBR056W, responsible for the hydrolysis of branched β-1,6-glucan, was downregulated. These findings suggest that strain FDHY23 reinforces its cell wall into a mesh-like structure to withstand high temperatures more effectively.

For glucose-6-P, the pentose phosphate pathway is its third downstream pathway (shown in the upper right part of Fig. 8). This pathway has four different directions. (1) Returning to glycolysis, the involved genes TKL1 and TAL1 were downregulated. (2) Proceeding from the intermediate product ribulose-5-P to synthesize FMN and pyridoxal-5-P, both of which were upregulated. FMN mediates mitochondrial respiratory chain, while pyridoxal-5-P is involved in amino acid deamination reactions, providing a nitrogen source for nucleotide synthesis. (3) Proceeding from ribose-5-P to generate ribose-1-P and PRPP, both of which were also upregulated, promoting nucleotide synthesis. (4) Proceeding from the intermediate product erythrose-4-P, together with the main glycolytic intermediate P-enolpyruvate, to generate aromatic amino acids and tetrahydrofolate (THF). Aromatic amino acids synthesis was downregulated, while THF generation was upregulated, promoting purine and pyrimidine synthesis. The fourth downstream pathway of glucose-6-P is inositol-3-P production, which was upregulated and may facilitate the synthesis of inositol phospholipids in the cell membrane.

As shown above, strain FDHY23 effectively reprogrammed carbon flux allocation at high temperature, by boosting the processes of lipid biogenesis, respiratory chain ATP generation, glycogen synthesis and nucleotide production, accompanied with elevated levels of intracellular NADPH and NAD+. FDHY23 not only enhanced glucose transport but also increased nitrogen uptake for material synthesis. Additionally, it enhanced vacuolar function and microautophagy to facilitate resource recycling (Supplementary Data 4). These adaptations enabled efficient resource utilization and optimized material synthesis at high temperatures. As a result, a large number of genes involved in cell cycle proliferation were upregulated in FDHY23. Additionally, some genes involved in cell mating and sporulation were upregulated, indicating a slight activation of sporulation as a response to high temperature. However, due to abundant ATP supply, rapid proliferation took priority over sporulation in FDHY23 at high temperatures.

### CYR1 mutation modulates the cAMP signaling pathway to rebuild global cellular processes under high temperature and recombinant protein production pressures

The CYR1 mutation directly decreases intracellular cAMP levels. Therefore, the detailed analysis below (Fig. 9) explores the impact of altered cAMP generation and its downstream cascades on overall cellular processes under conditions of high temperature and recombinant protein production.

For cAMP generation (Fig. 9), genes associated with mitochondrial function and the respiratory chain in FDHY23 were generally upregulated under high temperature and recombinant protein production conditions. This led to an increase in ATP production, which serves as the precursor substrate of cAMP. The enzyme CYR1, responsible for catalyzing cAMP generation, was also upregulated at high temperatures, while its inhibitory gene IRA2 showed downregulation. The genes PDE1 and PDE2, involved in cAMP degradation, did not exhibit significant change in expression. These findings indicate that FDHY23 upregulates CYR1 expression and increases

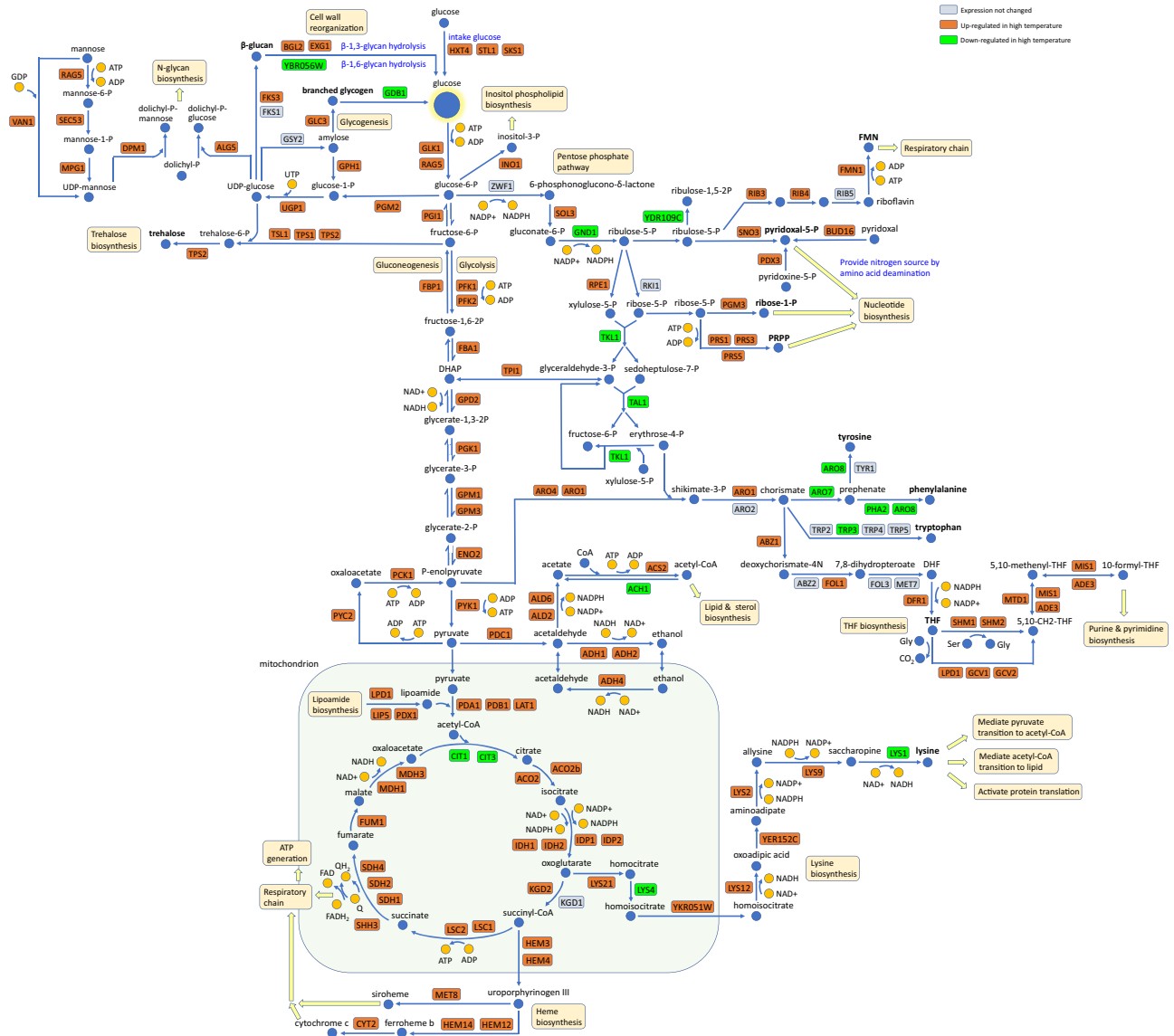

**Fig. 8 | The distribution map of differentially expressed genes involved in carbon metabolism pathways during logarithmic growth of FDHY23 at high temperatures.** LHP1044 serves as the control. The upregulated, downregulated, and unchanged genes are represented by the colors orange, green, and blue-gray, respectively.

ATP levels as a substrate for cAMP generation at high temperatures, resulting in a modest restoration of cAMP levels observed during mid-log phase of FDHY23 at high temperatures (Fig. 6d).

The protein kinase A (PKA) is the sole known downstream effector of cAMP in yeast. cAMP activates PKA by binding to its inhibitory subunit BCY1 (Fig. 9). The expression levels of PKA subunits in FDHY23 decreased under high temperature and recombinant protein production conditions (Fig. 9), consistent with the reduction in cAMP. In *S. cerevisiae*, repression of PKA activity promoted protein folding, stimulated gluconeogenesis and glycogen synthesis, inhibited glycolysis, enhanced mating and sporulation, and suppressed cell proliferation under heat stress[18–21]. In FDHY23 (Fig. 9), reducing PKA activity at high temperature improved protein folding and glycogen synthesis, similar to *S. cerevisiae*. However, unlike *S. cerevisiae*, glycolysis and cell proliferation were actually increased in FDHY23. This could be due to the heightened need for respiratory chain production, which upregulates glycolysis. Additionally, the ample supply of ATP may have fueled the upregulation of cell proliferation.

In addition, in *S. cerevisiae*, PKA activated the transcription factor RAP1, which in turn promoted the transcription of ribosomal genes through RAP1 binding sites their promoters[22]. In FDHY23, despite reduced PKA activity, most ribosomal genes were upregulated under high

temperature and recombinant protein production conditions (Supplementary Data 4). Meanwhile, *RAP1* expression remained unchanged, and its binding sites were only present in a few ribosomal genes. This indicates that the regulation of ribosomal gene transcription in FDHY23 may involve alternative mechanisms, rather than the PKA-RAP1 pathway. It has been reported that decreasing cAMP levels in *E.coli* can stimulate ribosome recovery[23]. Thus, the reduction in cAMP levels in FDHY23 may facilitate ribosome biogenesis through specific mechanisms, compensating for the loss caused by protein denaturation and degradation at high temperatures.

To identify new potential cAMP-regulated proteins in *K. marxianus*, we performed a conserved domain search on the entire genome. Based on the known cAMP-binding domains such as BCY1's CAP_ED domain and PDE2's GAF domain, through the "Conserved Domains" tool provided by NCBI, we identified three proteins in *K. marxianus* that contain known cAMP-binding domains. These include NTE1 and FBXL7, both containing the CAP_ED domain, and YKL069W, with the GAF domain. NTE1 is a phospholipase that converts phosphatidylcholine (PC) to glycerophosphocholine, playing a role in the rapid turnover of PC during temperature increase[24]. At high temperatures, *NTE1* was upregulated in FDHY23, along with the majority of genes involved in lipid and sterol synthesis.

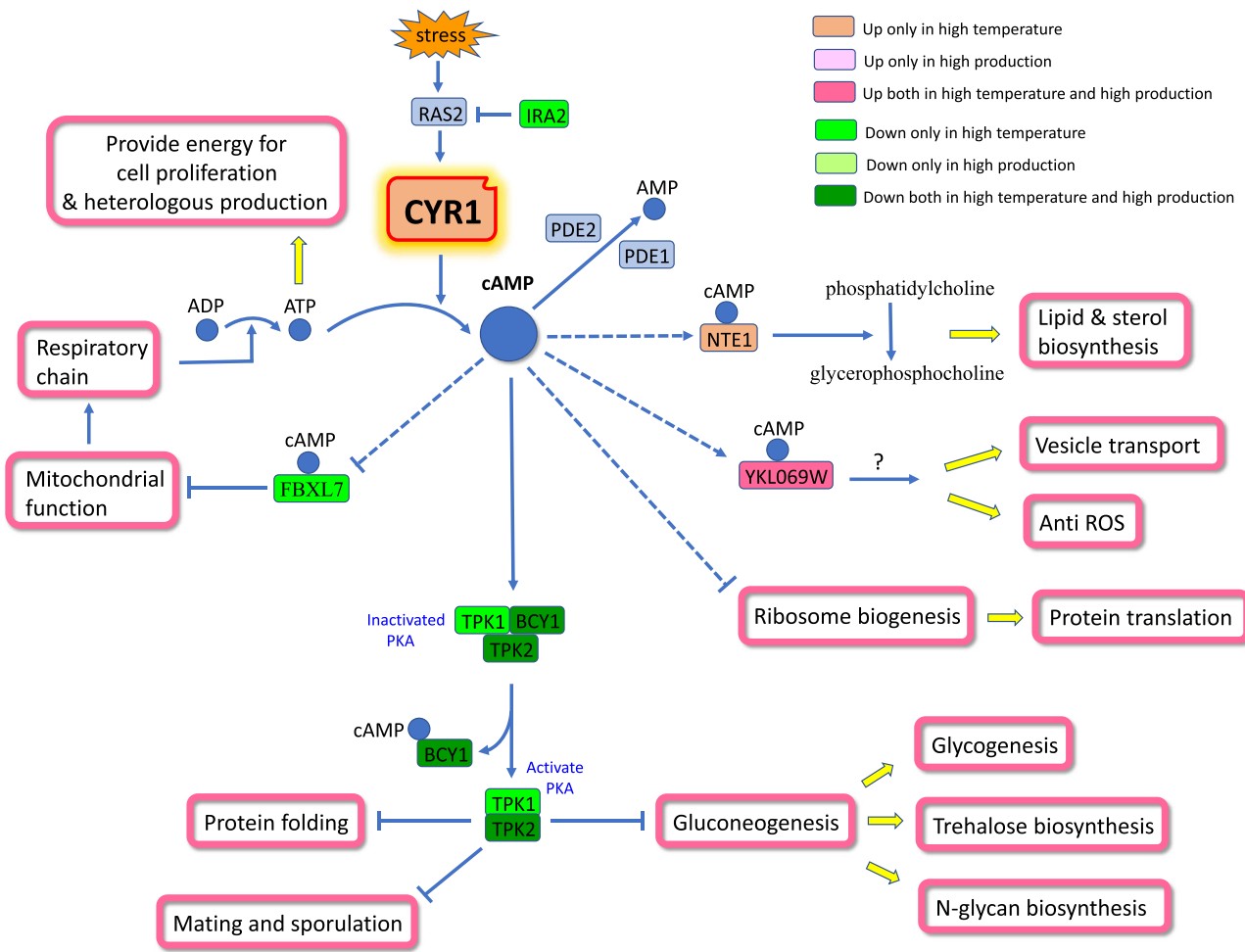

**Fig. 9 | The altered cAMP signaling cascades caused by the CYR1 mutation during high temperature and recombinant protein production conditions.** The color-gene differential expression correspondence is shown in the top right corner of the figure.

This suggests that the decrease in cAMP at high temperatures may stimulate the upregulation of its binding protein NTE1, leading to PC turnover as the membrane lipid signal to trigger lipid and sterol production. FBXL7 is an E3 ubiquitin ligase subunit and its deficiency has been shown to enhance the transcription of mitochondrial genes[25]. In *S. cerevisiae*, a brief pulse of cAMP led to a notable but temporary rise in mitochondrial transcript levels[26]. In FDHY23, *FBXL7* was downregulated at high temperatures (Fig. 9), while mitochondrial genes were predominantly upregulated. This indicates that at high temperature, *FBXL7* may mediate the promotion of mitochondrial gene transcription by transient cAMP pulses induced by respiratory chain augmentation. Gene *YKL069W*'s function is unknown, but it was upregulated in FDHY23 during both high temperature and recombinant protein production. Additionally, genes associated with Golgi vesicle transport and resistance to ROS were commonly upregulated in these conditions (Supplementary Data 4). Previous researches demonstrated that the addition of cAMP can enhance the expression of genes involved in vesicle transport at high temperatures[27] and boost cellular resistance to ROS[28]. Therefore, it is possible that *YKL069W* mediate the pulsatile increase in cAMP levels, leading to the upregulation of vesicle transport genes and anti-ROS genes.

In summary, the CYR1 mutation reduces cAMP levels, which in turn activates multiple cascade pathways during high temperature and recombinant protein production, including respiratory chain, lipid and sterol synthesis, vesicular transport, ROS resistance, protein folding, glycogen biosynthesis, and ribosome biogenesis. These changed cellular processes subsequently affect the allocation of metabolic resources within the cell.

## Discussion

Currently, engineering modifications of host cells to improve recombinant protein yields predominantly focus on enhancing protein folding and secretion process, with limited consideration for the overall stress sensitivity[4]. The use of low temperatures is so far the primary approach for utilizing environmental temperature stress to enhance recombinant protein production in yeast strains[4,29–32]. This involves reducing protein synthesis rates to allow for proper protein folding and suppressing protein degradation[33,34]. However, in industrial settings, high cell density fed-batch fermentations for yeast production of recombinant proteins generate significant heat. Increased temperatures, such as from 33 °C to 37 °C, are known to decrease recombinant protein yield[35,36]. Consequently, continuous cooling water addition is necessary to maintain optimal temperatures, resulting in substantial costs. Therefore, the pressing objective in industrial production is to enable yeast strains to produce high yields of recombinant proteins even at high temperatures.

However, in yeast cells, high-temperature response and growth viability often have a negative correlation. When the temperature exceeds the optimal range, genes involved in protein folding are induced[4], and cells enter a low metabolic state, accumulating glycogen and entering a G0 phase[37]. To achieve rapid cell growth and high yields of recombinant proteins under high temperature conditions, it is essential to enhance the strain's stress resistance and optimize their energy system to provide sufficient ATP. However, finding a breakthrough point to bridge the gap between stress resistance and the energy system remains challenging via rational design.

In this study, a *K. marxianus* mutant strain FDHY23 was screened under high temperature pressure. This strain showed improved growth

capacity at high temperature and increased expression levels of various recombinant proteins, attributed to the determinant mutation of CYR1[N1546K]. CYR1 catalyzes the generation of cAMP, an essential second messenger in cells under pressures, regulating a series of stress response pathways. Under high temperature conditions, a large amount of ROS is generated due to the heightened collisions of oxygen molecules[38]. This is also experienced by cells during recombinant protein production, where the intense folding of proteins leads to ROS accumulation. Therefore, ROS is a common and severe stress faced by cells during high temperature and recombinant protein production. Adding cAMP has been shown to eliminate ROS in *S. cerevisiae*[28], and also enhance cellular survival in plants at high temperatures by promoting the expression of superoxide dismutase (SOD)[27]. The CYR1[N1546K] mutation in FDHY23 reduces cAMP production, which is crucial for combating ROS during high temperature and recombinant protein production conditions. As a compensatory response to the decreased cAMP levels, there may be an increase in the generation of its substrate ATP. Consequently, the expressions of mitochondrial genes and respiratory chain genes were generally increased (Supplementary Data 4). This increased ATP generation may be the critical coupling point for the fast cell growth at high temperature and the high yields during recombinant protein production, as cell proliferation, protein folding and vesicle transport all heavily depend on ATP. Furthermore, the reduction in cAMP levels inactivates PKA, thereby promoting protein folding and glycogen synthesis. Altogether, the combined enhancement of ATP supply and stress resistance results in the improved thermotolerance and recombinant protein-producing capability.

In particular, the *CYR1* mutation is noted for its promoting effect on protein folding. In the context of protein folding, polypeptide chains spontaneously fold into secondary structures based on their amino acid sequences without the need for energy. However, in the process of forming tertiary structures, assistance from molecular chaperone systems is required, along with ATP to provide energy. Although the formation of disulfide bonds does not require energy, it generates a large amount of $H_2O_2$[39,40]. Our results have revealed that the *CYR1* mutation stimulates the upregulation of the molecular chaperone gene *HSP26* through the cAMP-PKA signaling pathway. This mutation also boosts intracellular ATP generation and resistance to ROS under high temperature and high production conditions, leading to the upregulation of genes related to disulfide bond formation such as *MPD2*, *FMO1*, and *ERO1*. Therefore, the *CYR1* mutation enhances the efficiency of protein folding by strengthening both the molecular chaperone system and the disulfide bond formation system. We analyzed the amino acid sequences (Supplementary Table 1) of the six produced recombinant proteins (LBA, HVP2, PCV2, Est1E, AnFaeA, and Badgla) in Fig. 3a–f using PROSITE for disulfide bond analysis[41], but no disulfide bonds were found. This suggests that highly expressed recombinant proteins in strain FDHY23 primarily benefit from the host cell's molecular chaperone system for protein folding.

It is worth mentioning that as early as 1984, a *CYR1* mutation was discovered in a *S. cerevisiae* strain, which resulted in a 1000-fold increase in thermotolerance and continuous chaperone expressions[42]. In 2000, Van Dijck et al. employed heat shock as a selection protocol and isolated a mutated *S. cerevisiae* strain with an amino acid mutation at position 1682 of the *CYR1* gene, located in the catalytic domain[43]. This mutation reduced enzyme activity and cAMP production, resulting in an improved resistance to freezing and drying conditions during fermentation when introduced into the *S. cerevisiae* strain[43]. Due to the significant generation of ROS in high temperature, freezing and drought environments[38,44], we further speculate that in stressful conditions that induce ROS production, a mutation in the catalytic domain of CYR1, resulting in reduced cAMP production, may trigger cells to upregulate mitochondrial function and respiratory chain ATP generation. This upregulation is aimed at restoring cAMP levels and counteracting ROS. Therefore, it offers a new target for rational engineering to enhance strain resilience and vitality.

In summary, our study provides a different perspective on cellular heat tolerance. Previous reports on yeast thermotolerance have primarily

concentrated on increasing the expression of heat shock responsive genes (e.g., molecular chaperones), and promoting trehalose production and cell membrane synthesis[45]. Our findings reveal that under heat stress, the *CYR1* mutation not only promotes protein folding, lipid synthesis, and glycogen production, but also enhances mitochondrial function and ATP generation. This provides the energy required for rapid cell growth alongside enhanced stress resistance. Additionally, we have shown that improving the host cell's thermotolerance can enhance its capacity for recombinant protein production. This suggests that thermotolerance can be utilized as an effective approach for high-throughput screening of challenging-to-detect high-yield recombinant proteins.

## Methods
### Strains and plasmids
The strains, plasmids, and primers used in this study are described in Supplementary Tables 2–4, respectively. The wild-type strain LHP1044 and the mutant strain FDHY23 were used in this study, with FDHY23 being obtained through high-temperature selection. A point mutation in a gene in *K. marxianus* was achieved using CRISPR/Cas9. LHZ531 was used as a backbone to build CRISPR plasmids[46]. The sgRNAs designed to *CYR1* were annealed in pairs and inserted into *Sap* I site of LHZ531 to obtain CRISPR plasmid 1925-3. The donor sequence was amplified from FDHY23 and co-transformed with CRISPR plasmid 1925-3 into LHP1044 to obtain strain LHP1044-CYR1[N1546K]. Similarly, the donor sequence was amplified from LHP1044 and co-transformed with CRISPR plasmid 1925-3 into FDHY23 to obtain strain FDHY23-CYR1. The gene-editing approach for point mutation of *LAP2* was similar to that used for the *CYR1* gene.

The pUKDN115 vector (without signal peptide sequence) and pUKDN112 (containing *INU* signal peptide sequence) vector were used for recombinant protein expression[46,47]. A series of recombinant protein expression plasmids were constructed for production in *K. marxianus*. The heterologous genes (*LBA*, *HVP2*, and *PCV2*) were expressed intracellularly using the pUKDN115 vector, while the heterologous genes (*Est1E*, *AnFaeA*, and *Badgla*) were expressed via secretion using the pUKDN112 vector.

### Screening of high-temperature tolerance
To screen for high-temperature tolerant strains, LHP1044 was inoculated in 50 mL of YPD medium (2% peptone, 2% glucose, and 1% yeast extract) and grown for 12 h at 30 °C. The suspension was then diluted with mQ water and spread onto YPD plates, which were subsequently cultured at 46 °C for 2 to 4 days. Colonies showing growth advantages were selected and grown at 46 °C in 3 mL of YPD medium for 4 h. Then, 5 μL of culture was spotted onto YPD plates and grown at 46 °C for 18 h and 36 h, respectively. Substantial growth on the plates indicates high-temperature tolerance.

### Estimation of high-temperature tolerance
For comparing the high-temperature tolerance of LHP1044 and FDHY23, both strains were inoculated in 50 mL of YPD medium (2% peptone, 2% glucose, and 1% yeast extract) and cultured at 30 °C for 12 h. For the spotting test, cells were collected and adjusted to an OD600 of 0.6. The cell suspension was then serially diluted fivefold, and 5 μL of each dilution was spotted onto YPD plates. The plates were subsequently incubated at 30 °C, 46 °C, 47 °C, and 48 °C, respectively. For the shaking test, cells were transferred to 50 mL of fresh YPD medium at an initial concentration of 0.2 OD600. The flasks were then shaken at different temperatures (30 °C, 46 °C, and 47 °C), and the absorbance values were recorded at intervals of 6 h or 12 h to measure the growth curves of LHP1044 and FHDY23. Each test was performed in biological triplicate.

### Recombinant protein assays and quantitation
Recombinant protein expression plasmids were transformed into *K. marxianus*. Transformants were inoculated in 50 mL of YD medium (4% glucose and 2% yeast extract) and cultured at 30 °C, 220 rpm for 72 h. The yields of desired proteins were measured by gray scanning or specific enzymatic activity. For quantifying the intracellular recombinant proteins,

the samples with the same $OD_{600}$ were harvested by centrifugation and the cell pellets were lysed using the previously reported method[47]. The supernatants of the cell lysates were resuspended in the SDS loading buffer and analyzed by SDS-PAGE. The absolute quantification of band intensity was estimated using GenoSens software by gray scanning analysis using different concentrations of β-lactoglobulin (Sigma, USA) as standards. The linear correlation between β-lactoglobulin concentration and grayscale values facilitates the conversion of grayscale values to the concentration of the target protein. For quantifying the extracellular recombinant proteins, the samples were harvested by centrifugation from the fermentation supernatant. The activities of Est1E, AnFaeA, and Badgla in the supernatant were measured as described previously[16,48].

### Adenylate cyclase assays

Adenylate cyclase activity was assayed in crude yeast plasma membranes using a method described previously[49]. Crude membranes were prepared according to the established protocol. The assay was performed in a reaction mixture consisting of 20 mM MES (pH 6.2), 2.5 mM $MgCl_2$, 0.1 mM EGTA, 1 mM 2-mercaptoethanol, 1 mM DTT, and 1 mM ATP. The reaction was initiated by adding the reaction mixture to the membranes in a final volume of 100 μl. The tubes were then incubated at 30 °C for 60 min and terminated by boiling for 5 min. After boiling, the samples were mixed with an equal volume of methanol. The mixture was then centrifuged and the supernatant was passed through a 0.22 um filter. The cAMP produced was determined by HPLC. The protein concentration was measured by the BCA Protein Assay Kit (Sangon Biotech, Shanghai, China). The enzymatic activity unit (U) of adenylate cyclase was defined as the amount of cAMP (pmol) produced by 1 mg protein per minute at 30 °C.

### Quantification of cAMP by HPLC

The concentrations of cAMP were measured using HPLC as described by Matencio A with minor modifications[50]. 20 μL of the sample was injected into an Agilent 1200 series HPLC system (CA, USA) equipped with a 1200 series module UV-VIS detector and a Zorbax Eclipse Plus C18 column (250 mm × 4.6 mm, 5 μm particle size) at 30 °C. The HPLC conditions were as follows: flow rate at 1 mL/min, mobile phase A consisted of mQ water with 0.1% acetic acid, mobile phase B consisted of 85/15 (w/w) MeOH/THF with 0.1% acetic acid. Gradient: 0–3 min, 0% B; 3–10 min, B from 0% to 5%; 10–20 min, B from 5% to 50%; 20–25 min, B from 50% to 0%; 25–30 min, 0%B. The UV detector was operated at 340 nm.

### High-density fermentation

The fresh transformants of LHP1044-LBA and FDHY23-LBA strains were inoculated in 150 mL SD medium[51] and grown at 30 °C, 220 rpm for 18 h, respectively. The culture was then transferred into a fermentor containing 2 L of defined mineral medium[52]. High-density fermentation was carried out in a 5 L fermentor (BXBIO, Shanghai, China) as described recently[52]. During fermentation, the dissolved oxygen was maintained at 5–15%, temperature was controlled at 32 °C, and pH was adjusted to 5.5 using ammonia water. Samples were collected every 12 h and diluted tenfold for analysis of LBA yield by SDS-PAGE, as described above. The absorbance values were recorded at intervals of 12 h to measure the growth curves. Three separate batches of fermentation were conducted as biological replicates.

### RT-qPCR analysis

The relative expressions of genes were determined by real-time quantitative reverse transcription PCR (RT-qPCR). Yeast cells were grown to indicated phase and harvested by centrifugation. The total RNA was extracted using quick-RNA fungal/bacterial miniprep kit (ZYMO RESEARCH, China) and then the cDNA was generated by HiScript III All-in-one RT SuperMix Perfect for qPCR (Vazyme, China). The generated cDNA was used as RT-qPCR templates. RT-qPCR was conducted on an LightCycler® 480 Instrument II (Roche, Switzerland) using ChamQ Universal SYBR qPCR

Master Mix (Vazyme, China). The house-keeping gene SWC4 was used as the reference gene. The primers for RT-qPCR were listed in Supplementary Table 5.

### DNA sequencing

Strains LHP1044 and FDHY23 were inoculated in 50 mL of YPD medium at 30 °C, respectively. Overnight-cultivated cells were collected and washed twice with per-chilled mQ water. Then, the samples were sent to Sangon Biotech (Shanghai, China) for DNA sequencing. The reads were mapped to the reference genome of *K.marxianus* FIM1 (GenBank assembly accession: GCA_001854445.2). The average coverage is 247× and 393× for the LHP1044 and FDHY23, respectively, and then the SNPs or InDelS in FDHY23 against LHP1044 were identified.

The sequenced genome of the K. marxianus FIM1 strain[14] (https://www.ncbi.nlm.nih.gov/datasets/genome/GCA_001854445.2/), which did not include the mitochondrial genome at that time, served as the reference genome for nuclear genes. The mitochondrial genome of the K. marxianus DMKU3-1042 strain[53] (https://www.ncbi.nlm.nih.gov/datasets/genome/GCF_001417885.1/) was used as the reference genome for mitochondrial analyses.

### RNA sequencing

Strains LHP1044 and FDHY23 were inoculated in 50 mL YPD medium at 30 °C, respectively. Overnight-cultivated cells were transferred to 50 mL fresh YPD medium. LHP1044 and FDHY23 were cultured at 46 °C for 48 h and 24 h, respectively, to ensure cells reaching the mid-log phase. Afterwards, cells were collected and washed twice with per-chilled mQ water. Then the cell pellets were stored at −80 °C until further analysis. LHP1044-LBA and FDHY23-LBA were inoculated in 50 mL SD medium and grown overnight at 30 °C, respectively. The cells were transferred to 50 mL YD medium and cultured for 60 h. Similarly, cells were collected and washed twice with per-chilled mQ water. Then the cell pellets were stored at -80 °C before analysis. These samples were sent to Biozeron (Shanghai, China) for RNA-seq ananlysis. The differential expression of genes between two samples was analyzed using edgeR (v3.8) or *DESeq2*. Genes with | $\log_2 FoldChange| >= 1$ and p-values < 0.05 are defined as significantly differentially expressed. Samples for RNA-seq investigation were performed in biological triplicate.

### Statistics and reproducibility

All experiments in this study were conducted with a minimum of three biological replicates (except the DNA sequencing). Student's t-test was used for inter-group sample significance analysis, with a p-value less than 0.05 indicating significant differences between the two groups. Cohen's d value was utilized as the statistical metric for measuring effect size. In the results, '*' represents $p < 0.05$, '**' represents $p < 0.01$, and '***' represents $p < 0.001$. The n biological replicates are defined as follows: after strain activation, n single clones were selected and inoculated into n independent tubes or shake flasks for separate experiments, where n represents the number of biological replicates. Error bars in each figure are calculated as the standard deviation (SD).

### Reporting summary

Further information on research design is available in the Nature Portfolio Reporting Summary linked to this article.

## Data availability

The original DNAseq and RNA-seq datasets presented in this study can be found in online NCBI Sequence Read Archive (SRA) repositories as follows: https://www.ncbi.nlm.nih.gov/bioproject/PRJNA1100266. The numerical source data behind the graphs can be found in Supplementary Data 5. All untrimmed and unedited blot/gel images are provided as Supplementary Figures in the Supplementary Information file. All other data are available from the corresponding author on reasonable request.

## Code availability
The DNAseq analysis were carried out by Sangon Biotech (Shanghai, China). The differential expression of genes between two sample groups was analyzed using edgeR (https://www.bioconductor.org/packages/release/bioc/html/edgeR.html).

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

## Acknowledgements

We would like to acknowledge the colleagues in Engineering Research Center of Industrial Microorganisms in Fudan University for their generous assistance. This work was supported by the National Key Research and Development Program of China (Nos. 2021YFC2100203 and 2021YFA 0910601), and the Science and Technology Research Program of Shanghai (No. 2023ZX01).

## Author contributions

H.L. and W.M. supervised the study. H.L. and H.R. designed the experiments. H.R. performed most of the experiments. Q.L. optimized the quantitative method of cAMP by HPLC. H.R., W.M., H.L., S.Z. and Y.L. analyzed the data. H.R., W.M., Y.Y. and J.Z. interpreted the results. W.M. and H.R. wrote the manuscript. H.L. organized this research project. All authors read and approved the final draft of the manuscript.

## Competing interests

The authors declare no competing interests.
