## [Peer Review File · Communications Biology]

Reviewers' comments:

Reviewer #1 (Remarks to the Author):

Comments on manuscript titled " Coupling thermotolerance and high production of recombinant protein by CYR1N1546K mutation via cAMP signaling cascades"

In the present study, authors screen for a thermotolerant strain which also improved expression of a recombinant protein at both optimal and high temperature. Authors further identify CYR1 mutant as the basis of this observed phenotype. The cAMP signalling is affected in the strain resulting in reduced cAMP which is shown as a basis of observed thermotolerance. Data are convincing, controls and statistical analysis are appropriate. I have following comments:

- (1) Figure 4C: How was protein concentration estimated?
- (2) Does the evolve strain has any effect on folded state of the protein especially for proteins containing disulfide bonds. At least one of the proteins could be purified and secondary state examined using methods such as CD.
- (3) Have authors also examined the mitochondrial encoded genome? This should be mentioned in the text.
- (4) Line 193-195: " The above results indicatecompared to LHP1044". Authors mention that FDHY23 enhances cAMP level at high temperature however the data shows that the level are not much affected and infact there is a slight drop.
- (5) Authors show that the CYR1 mutation is responsible for thermotolerance by introducing the same mutation in LHP1044 strain (LHP1044-CYR1(N1546K)). Authors should also use this strain (LHP1044-CYR1(N1546K)) to show that the enhanced protein expression is solely due to this mutation.

Reviewer #2 (Remarks to the Author):

This manuscript describes the identification and verification of a yeast strain that produces a recombinant protein at high yields using thermo-adaptation. The work presented is novel and the results very exciting. There are a few issues that need to be addressed further, as listed below. I recommend that this article be accepted with major revision.

1. Figure 1C: This plot is actually unnecessary. The highest OD600 reached by each strain under each cultivation temperature is clearly presented in Figure 1B. In addition, the authors use "biomass" for the Y-axis, then it is recommended to use dry cell weight instead of OD600 to reflect the actual biomass accumulation.
2. Figure 2B: First, there is too much redundant information. The five strains tested can be tested in a single plate and shown in a single picture, without showing the LHP1044 and FDHY23 repetitively. Second, the resolution of these pictures is too low. Third, in the second spot picture, the authors use "HY23" instead of "FDHY23" to indicate the strain. I guess this is a typo.
3. Line 125: "room temperature" usually indicates 20-25 degrees, but here the authors use 30 degrees. So the wording needs to be modified correspondingly.
4. Figure 3: Please indicate in the Methods section the normalization method used for SDS-PAGE. Did the authors use the same amount of cells for gel analysis of different samples or the same amount of total protein?
5. Figure 4B: Please add the error bar of several independent experiments, i.e., data derived from several batches of fermentation.
6. Results section 4 is about the protein expression at 46 degrees, but the 5-L fermentation was performed at 32 degrees as mentioned in the Methods section.
7. Line 163: The authors stated that "the CYR1 N1546K mutation has demonstrated the ability to

significantly enhance recombinant protein production". However, strain FDHY23 contains many mutations compared with LHP1044, with CYR1 N1546K being a major one. So, it is incorrect to say that the higher expression level at 46 degrees is a result of this mutation alone. It is suggested that this mutation be introduced into strain LHP1044, as conducted in Figure 3H, to further investigate how important this mutation is to the LBA protein expression at a higher temperature.

8. Results section 5 and Figure 5: When the authors mention "logarithmic phase", please indicate if this is the early-, mid-, or late-log phase.

9. Lines 191-193: What is the "platform period" of LBA production? There is no corresponding data, and I cannot get the relevant information from the gel pictures.

10. Figure 5D and E: What are the levels of cAMP in these strains at the stationary phase?

11. Lines 193-195: This conclusion cannot be made without the data on the absolute value of cAMP levels in various strains.

12. Lines 200-205: For RNA-seq analysis, why did the authors choose the log-phase for the high-temperature condition while choosing the stationary phase for the high-production condition? It appears that the LBA expression level is not the highest at the stationary phase. For the high-production condition, cells are supposed to be stressed to the most extent in the log-phase as the highest amount of LBA is produced.

13. Figure 6: "up-regulation" and "down-regulation" should be "up-regulation" and "down-regulation", respectively.

14. Based on the RNA-seq data, the authors analyzed several key genes and pathways that could contribute to the observed phenotype. It is suggested that qRT-PCR or western blot be performed for the key genes/enzymes to confirm the change in their expression levels.

The comments and suggestions provided by the two reviewers are very helpful, leading to significant improvement in the manuscript. The revised parts have been highlighted in blue throughout the manuscript. A detailed point-to-point response is provided below to address the concerns raised by the reviewers.

Reviewer number: 1

1. Figure 4C: How was protein concentration estimated?

Response:

The protein concentration in previously mentioned Figure 4C (now referred to as Fig.4B and Fig.4D in the revised manuscript) was estimated using a method described in the 'Recombinant protein assays and quantitation' section of the Methods. The absolute quantification of band intensity was estimated using GenoSens software by gray scanning analysis using different concentrations of β -lactoglobulin (Sigma, USA) as standards. The linear correlation between β -lactoglobulin concentration and grayscale values facilitates the conversion of grayscale values to the concentration of the target protein. The corresponding SDS-PAGE analysis for absolute quantification is provided in the figure of Suppl. file 2.

2. Does the evolve strain has any effect on folded state of the protein especially for proteins containing disulfide bonds. At least one of the proteins could be purified and secondary state examined using methods such as CD.

Response:

Thanks very much for the reviewer's attention and professional advice on protein folding. Currently, our laboratory lacks the capability to observe protein folding status, such as using circular dichroism (CD) to detect protein secondary structure. However, we can infer an enhanced protein folding capacity in the mutant strain FDHY23 based on gene expressions. For instance, the chaperone protein HSP26, involved in tertiary structure formation, was upregulated in both high temperature and recombinant protein production conditions. Similarly, genes MPD2 and FMO1, involved in disulfide bond formation in the endoplasmic reticulum (ER) were upregulated in FDHY23 at 46°C. The gene ERO1, also involved in disulfide bond formation, was upregulated in FDHY23 during high production of recombinant protein LBA.

In terms of protein folding, the polypeptide chain spontaneously folds into specific secondary structures based on its amino acid sequence without the need for ATP energy involvement. However, the formation of tertiary structure requires assistance from molecular chaperone systems such as the HSP70 family, J protein family, and nucleotide exchange factor NEF. During the assisted folding process, ATP bound to HSP70 is hydrolyzed to ADP, releasing energy and promoting protein folding. Although disulfide bond

formation does not directly require ATP, it depends on O₂ for oxidation and generates a significant amount of H₂O₂ (Sevier, 2006; Patil, 2015).

In our study, through RNA-seq data analysis, we found that CYR1 mutation enhances respiratory chain, ATP generation, ROS resistance, and disulfide bond formation under high temperature and recombinant protein production conditions. Furthermore, molecular chaperone genes were upregulated through the cAMP-PKA cascade pathway. These findings suggest that the CYR1 mutation promotes protein folding efficiency by strengthening both the molecular chaperone system and disulfide bond system.

Regarding the highly expressed recombinant proteins (LBA, HVP2, PCV2, Est1E, AnFaeA, and Badgla) in strain FDHY23 mentioned in the revised manuscript (Fig. 3A ~ 3F), their amino acid sequences (Suppl. file 6_Table S1) were analyzed for disulfide bonds using PROSITE (<http://prosite.expasy.org>) (Sigrist, 2013). However, no disulfide bonds were predicted in these sequences. This suggests that the enhanced expression of these recombinant proteins in FDHY23 primarily benefits from the host cell's abundant molecular chaperone system and ATP for promoting tertiary structure formation.

We have incorporated this protein folding analysis into the discussion section of the revised manuscript.

Reference:

N. A. Patil, et al., Cellular disulfide bond formation in bioactive peptides and proteins. *Int. J. Mol. Sci.*, 2015, 16:1791-1805.

C. S. Sevier, et al., Conservation and diversity of the cellular disulfide bond formation pathway, *Antioxidants & Redox Signaling*, 2006, 8:5 & 6.

C. J. A. Sigrist, et al., New and continuing developments at PROSITE, *Nucleic Acids Res.* 2013, 41:D344-7.

3. Have authors also examined the mitochondrial encoded genome? This should be mentioned in the text.

Response:

Thanks a lot for the reminding. In this study, the KM_FIM1 strain is used, so when analyzing DNA-seq and RNA-seq data, we employed the previously sequenced KM_FIM1 genome (Yu, 2021) as the reference genome. However, it should be noted that this reference genome does not include the mitochondrial genome. Fortunately, the KM_DMKU genome (Lertwattanasakul, 2015) does contain the mitochondrial genome. Therefore, in this revised manuscript, we have reanalyzed the DNA-seq and RNA-seq data for mitochondrial-encoded genes using the mitochondrial genome of KM_DMKU (NC_036032.1) as the reference genome.

We found that the mutant strain FDHY23 showed only two SNP mutations (T

→G at 46283, T→A at 46287) in the intergenic region of the mitochondrial genome, both occurring after the last coding gene (tRNA, 44526~44598), when compared to the wild-type strain LHP1044. Thus, there are no discernible functional differences in the mitochondrial genome between the two strains. The detailed analysis of mitochondrial mutations is provided in Supplementary file 1_TableS2. We have now included this mitochondrial mutation analysis in Section 2 of the Results and in the 'DNA sequencing' section of the Methods.

To analyze the differential expression of mitochondrial-encoded genes, we mapped the RNA-seq data to the KM_DMKU mitochondrial genome. We then counted the number of reads mapped to mitochondrial coding genes and combined these counts with the total count of reads mapped to nuclear genes in the KM_FIM1 genome. This allowed us to calculate the FPKM values, p-values, and adjusted p-values for the mitochondrial-encoded genes, as shown in the last eight rows of Suppl. file 2 and Suppl. file 3.

No significant differences in the expression of mitochondrial-encoded genes were found between FDHY23 and LHP1044. However, we did note a slight upregulation of COX1 and two rRNA genes in FDHY23 under high temperature condition. It is important to note that nuclear-encoded mitochondrial genes, including mitochondrial ribosomal genes, respiratory chain genes, ATP generation genes, were significantly upregulated in FDHY23 under high temperature and recombinant LBA production conditions (Fig.7A in the revised manuscript). This suggests that nuclear-encoded mitochondrial genes play a more significant role in enhancing mitochondrial function. We have incorporated the analysis of mitochondrial expression into Section 6 of the Results.

Reference:

N. Lertwattanasakul, et al, Genetic basis of the highly efficient yeast *Kluyveromyces marxianus*: complete genome sequence and transcriptome analyses, *Biotechnol Biofuels*, 2015, 8:47.

Y. Yu, et al. Comparative Genomic and Transcriptomic Analysis Reveals Specific Features of Gene Regulation in *Kluyveromyces marxianus*. *Front. Microbiol.*, 2021, 12:598060.

4. Line 193-195: "The above results indicatecompared to LHP1044". Authors mention that FDHY23 enhances cAMP level at high temperature however the data shows that the level are not much affected and in fact there is a slight drop.

Response:

In the previous manuscript, we did not further compare the analysis results, which made it difficult to clearly draw conclusions. At 30°C, we observed that

the CYR1 activity (Fig. 6B in the revised manuscript) and the cAMP content (Fig. 6C in the revised manuscript) in the mutant strain FDHY23 were significantly lower than those in LHP1044 during both the logarithmic and stationary phases. Since the main difference between FDHY23 and LHP1044 at high temperatures is observed in the logarithmic phase (Fig. 1B), we mainly focused on comparing the cAMP content of the two strains during the logarithmic growth phase at 46°C (Fig. 6D in the revised manuscript). We found that the cAMP content of LHP1044 in the mid-log phase at 46°C was significantly lower compared to that at 30°C ($p = 0.018$), while the cAMP content of FDHY23 in the mid-log phase at 46°C was slightly higher than that at 30°C ($p = 0.213$). Therefore, we concluded that 'The above results indicate that FDHY23 enhances cAMP production during the logarithmic growth phase at high temperatures compared to LHP1044.' In the revised manuscript, in Section 5 of Results, we have modified this conclusion to be clearer as 'These results suggest that with the increase in temperature, FDHY23 enhances the production of cAMP during mid-log phase, while LHP1044 significantly drops its cAMP production.'

5. Authors show that the CYR1 mutation is responsible for thermotolerance by introducing the same mutation in LHP1044 strain (LHP1044-CYR1(N1546K)). Authors should also use this strain (LHP1044-CYR1(N1546K)) to show that the enhanced protein expression is solely due to this mutation.

Response:

In our initial experiments, we also examined the impact of another LAP2^{G353R} mutation in strain FDHY23 on the production of recombinant proteins. Specifically, we examined the LHP1044-LAP2^{G353R} strain, which was created by introducing the LAP2 mutation into the wild-type strain LHP1044. This mutant strain did not exhibit any increase in the yield of recombinant protein LBA compared to LHP1044. Although this result was not included in the previous manuscript, it has now been added to Fig. 3H in the revised manuscript. Additionally, the corresponding text has been incorporated into Section 3 of the Results. This finding further supports the notion that the CYR1N1546K mutation is also responsible for the enhanced production of recombinant proteins.

Reviewer number: 2

1. Figure 1C: This plot is actually unnecessary. The highest OD600 reached by each strain under each cultivation temperature is clearly presented in Figure 1B. In addition, the authors use "biomass" for the Y-axis, then it is recommended to use dry cell weight instead of OD600 to reflect the actual biomass accumulation.

Response:

Thanks a lot for this suggestion. We agree that Fig. 1C is redundant as the values it presents are already included in Fig. 1B. Therefore, in the revised manuscript, we have removed the previous Fig.1C and renamed the maximum growth rate graph as Fig. 1C.

2. Figure 2B: First, there is too much redundant information. The five strains tested can be tested in a single plate and shown in a single picture, without showing the LHP1044 and FDHY23 repetitively. Second, the resolution of these pictures is too low. Third, in the second spot picture, the authors use “HY23” instead of “FDHY23” to indicate the strain. I guess this is a typo.

Response:

We acknowledge that there was an excessive amount of repetitive information in the previous version of Fig. 2B and that it lacked conciseness. In response, we have reorganized the experiments in the revised Fig. 2B, ensuring that LHP1044 and FDHY23 are only presented once. Additionally, we have taken new photographs of the re-spotted data, hoping to meet the clarity requirements. We greatly appreciate the reviewer’s attention to the typo, and we have taken extra care to correct all typos in the revised manuscript.

3. Line 125: “room temperature” usually indicates 20-25 degrees, but here the authors use 30 degrees. So the wording needs to be modified correspondingly.

Response:

Thanks a lot for the meticulousness. We have confirmed that “room temperature” refers to the typical temperature range comfortable for human habitation indoors. It was inappropriate to use “room temperature” to describe 30°C. In the revised manuscript, we have rectified this error by replacing all instances of “room temperature” with “30°C”.

4. Figure 3: Please indicate in the Methods section the normalization method used for SDS-PAGE. Did the authors use the same amount of cells for gel analysis of different samples or the same amount of total protein?

Response:

We standardized the cell number using the sample with the same OD600, followed by further SDS-PAGE analysis. To provide clarity, we have included the following details in the ‘Recombinant protein assays and quantitation’ section of Methods: “For quantifying the intracellular proteins, the samples with the same OD600 were harvested by centrifugation and the cell pellets were lysed”.

5. Figure 4B: Please add the error bar of several independent experiments, i.e.,

data derived from several batches of fermentation.

Response:

We have conducted additional fermenter experiments at 32°C to confirm the significant increase in LBA production by strain FDHY23. In the previous version of the manuscript, we did not include these repeated fermenter data. In the revised manuscript, we have added two independent experiments conducted in fermenters at 32°C, in addition to the existing fermenter data. This provides a total of three separate batches of fermentation as biological replicates. Consequently, bar values have been added to the growth curve of LBA production in fermenters in Fig. 4C in this revision.

6. Results section 4 is about the protein expression at 46 degrees, but the 5-L fermentation was performed at 32 degrees as mentioned in the Methods section.

Response:

The 5-L fermentation is typically carried out at 32°C, and therefore it should not be included in the section for protein production at high temperatures. In the revised manuscript, we have relocated the data on LBA production in fermenters to Section 3 of the Results and included it in Fig. 4C and 4D.

7. Line 163: The authors stated that “the CYR1 N1546K mutation has demonstrated the ability to significantly enhance recombinant protein production”. However, strain FDHY23 contains many mutations compared with LHP1044, with CYR1 N1546K being a major one. So, it is incorrect to say that the higher expression level at 46 degrees is a result of this mutation alone. It is suggested that this mutation be introduced into strain LHP1044, as conducted in Figure 3H, to further investigate how important this mutation is to the LBA protein expression at a higher temperature.

Response:

To confirm that the enhanced protein production at high temperatures is attributed to the CYR1^{N1546K} mutation, we first provided additional data demonstrating that introducing the LAP2 mutation did not increase LBA production at 30°C (Fig. 3H in the revised manuscript). In contrast, introducing the CYR1 mutation into LHP1044 led to a remarkable increase in LBA production at 30°C (Fig. 3H). Furthermore, we evaluated LBA production at 46°C and observed that the strain with CYR1^{N1546K} mutation still exhibited higher levels of LBA protein compared to LHP1044 (Fig. 5C). These findings confirm that the introduction of the CYR1^{N1546K} mutation indeed enhances protein production at high temperatures.

8. Results section 5 and Figure 5: When the authors mention “logarithmic

phase”, please indicate if this is the early-, mid-, or late-log phase.

Response:

When sampling to measure cAMP content of the strains at high and normal temperatures, we determined our sampling time based on the growth curves. Due to the growth curves (Fig. 1B), the term "logarithmic phase" refers to the mid-log phase. In this revised manuscript, we have replaced all instances of "logarithmic phase" with "mid-log phase" in Sections 5 to 8 in the Results for better clarity.

9. Lines 191-193: What is the “platform period” of LBA production? There is no corresponding data, and I cannot get the relevant information from the gel pictures.

Response:

The “platform period” for LBA production by the strains refers to the stationary growth phase in the growth curves (Fig. 4A in the revised submission), specifically after 48 h. From the corresponding bands of LBA detection at different time points (Fig. 4B), it is evident that FDHY23 and LFP1044 both exhibited an increase in LBA production over time, with a noticeable surge at the onset of the stationary phase. In the revised manuscript, we have replaced “platform period” with “stationary phase” for consistency.

10. Figure 5D and E: What are the levels of cAMP in these strains at the stationary phase?

Response:

In our previous manuscript, we used the same sample of RNA-seq to detect the cAMP content of strains LHP1044 and FDHY23 during high-temperature growth, as well as during LBA production at 30°C. The sample was split into two portions—one for RNA-seq analysis and the other for cAMP measurement. As shown in Fig. 1B, the OD600 of both strains FDHY23 and LHP1044 during the stationary phase at 46°C did not show significant differences, while the growth discrepancy was most pronounced during the mid-log phase. Therefore, when comparing strains under high-temperature condition, the sampling was conducted during the mid-log phase rather than the stationary phase.

For LBA production at 30°C, their growth curves (Fig. 4A) of strains FDHY23 and LHP1044 did not exhibit significant differences, and their respective LBA yields (Fig. 4B) both reached a maximum during the stationary growth phase, with FDHY23's LBA yield notably higher than that of LHP1044. Therefore, for LBA production at 30°C, both RNA-seq and cAMP measurements were taken during the stationary phase. Consequently, in the revised manuscript, Fig.6D presents cAMP content in the mid-phase of strains FDHY23 and LHP1044 during high-temperature growth, whereas Fig. 6E depicts their cAMP content during the stationary growth phase for LBA production at 30°C.

In the revised manuscript, we also measured the cAMP content of strains FDHY23 and LHP1044 during the stationary phase at 46°C, as shown in Fig. 6D. It can be observed that during the mid-log phase at high temperatures, there is no significant difference in cAMP content between FDHY23 and LHP1044. However, during the stationary phase, the cAMP content in FDHY23 is significantly lower than that in LHP1044.

11. Lines 193-195: This conclusion cannot be made without the data on the absolute value of cAMP levels in various strains.

Response:

In this revised manuscript, we have provided the absolute values of cAMP content (Fig. 6C, 6D, and 6E). Upon comparison, it was found that the cAMP content of LHP1044 during the mid-log phase at 46°C is significantly lower than that at 30°C ($p = 0.018$), while FDHY23 shows slightly higher cAMP content during the mid-log phase at 46°C compared to 30°C ($p = 0.213$). Therefore, we conclude that “The above results indicate that FDHY23 enhances cAMP production during the logarithmic growth phase at high temperatures compared to LHP1044”. In the revised manuscript, we have modified the conclusion to make it clearer, as follows: “These results suggest that with the increase in temperature, FDHY23 enhances the production of cAMP during the mid-log phase, while LHP1044 significantly decreases its cAMP production”.

12. Lines 200-205: For RNA-seq analysis, why did the authors choose the log-phase for the high-temperature condition while choosing the stationary phase for the high-production condition? It appears that the LBA expression level is not the highest at the stationary phase. For the high-production condition, cells are supposed to be stressed to the most extent in the log-phase as the highest amount of LBA is produced.

Response:

Based on the growth curves of strains FDHY23 and LHP1044 at 46°C (Fig. 1B), it is evident that there is no significant difference in OD600 during the stationary phase, while the growth discrepancy is most pronounced during the mid-log phase. Therefore, for the purpose of comparing the strains under high temperature condition, sampling was conducted specifically during the mid-log phase.

For the LBA production of strains FDHY23 and LHP1044 at 30°C, their growth curves (Fig. 4A) did not show any significant differences. However, their respective LBA yields reached their maximum during the stationary growth phase at 72 h (Fig. 4B), with FDHY23 showing significantly higher LBA production compared to LHP1044. Thus, for the comparison of strains under recombinant protein production condition, sampling was taken during the

stationary growth phase.

From Fig. 4B, it can be observed that both strains FDHY23 and LHP1044 exhibit low LBA production during the log-phase, and the production of LBA significantly increases in the stationary phase after 48 h. Achieving a balance between high recombinant protein yield and cell proliferation is a challenging task. How cells effectively allocate resources between these two competing processes remains a complex problem. We are currently developing a digital whole-cell model to address this issue.

13. Figure 6: “up-regulation” and “down-regulation” should be “up-regulation” and “down-regulation”, respectively.

Response:

Thanks very much for the carefulness. In this revised manuscript, we have corrected the typo in Fig. 7A and 7B.

14. Based on the RNA-seq data, the authors analyzed several key genes and pathways that could contribute to the observed phenotype. It is suggested that qRT-PCR or western blot be performed for the key genes/enzymes to confirm the change in their expression levels.

Response:

Thanks a lot for the pertinent suggestion. In this revision, we have included the qRT-PCR data for 25 key genes (Fig. 7C). These genes encompass three involved in respiratory chain and ATP production (ATP23, CYC1, CBS2), three related to protein folding (HSP26, EMC6, DNAJA1), one participating in anti-ROS process (INP1), two contributing to ribosome biogenesis (MRPL17 and RPP1A), two genes associated with protein translation (CLU1 and IFM1), six involved in secretory pathway and vesicle transport (COG4, SEC62, SEC11, YFH7, SSA1, SSA3), four related to central carbon metabolism (FBP1, PDA1, FMN1, HEM3), and four associated with the cAMP cascade pathway (CYR1, BCY1, YKL069W, NTE1). The qRT-PCR results for most of these genes are consistent with the RNA-seq findings.

REVIEWERS' COMMENTS:

Reviewer #1 (Remarks to the Author):

The concern raised earlier has been answered. The manuscript is now acceptable for publication.

Reviewer #2 (Remarks to the Author):

This manuscript has been greatly approved, and the overall work is novel, interesting, and complete. There are only a few minor points that need to be addressed further.

1. Line 96: When mentioning "growth rate", did the authors refer to the specific growth rate? If so, please use the correct, complete term.

2. Figure 3, panes D and E: For the title of the Y-axis, there should be a space between "the" and "activity".

3. Line 203-204: "In contrast, the cAMP level of FDHY23 at 46°C was obviously higher than its cAMP level at 30°C ($p = 0.213$)."

The p value is more than 0.05, so the difference is insignificant. That is to say, there is no reliable difference between the two conditions. Also, in Lines 206-208, the authors state that "These results suggest that with the increase in temperature, FDHY23 enhances the production of cAMP during mid-log phase", which is incorrect.

4. There are a few grammatical errors:

(1) Line 27: "CYR1 mutation induced reduction in cAMP levels" should be "CYR1 mutation-induced reduction in cAMP levels".

(2) Line 55: "gene expressions" should be "gene expression".

(3) Line 65: "researches" should be "research".

(4) Line 175: "remarkable higher" should be "remarkably higher".

We are very grateful for the suggestions provided by the reviewers, and below is a detailed point-to-point response addressing these concerns.

Reviewer number: 1

The concern raised earlier has been answered. The manuscript is now acceptable for publication.

Reviewer number: 2

1. Line 96: When mentioning “growth rate”, did the authors refer to the specific growth rate? If so, please use the correct, complete term.

Response:

Thanks a lot for the reminding. The term “growth rate” in the manuscript indeed referred to the specific growth rate. We have completed this term in the revised manuscript, which is highlighted in blue.

2. Figure 3, panes D and E: For the title of the Y-axis, there should be a space between “the” and “activity”.

Response:

Thanks a lot for the carefulness. We have corrected the title of the Y-axis in Figure 3 in this revision.

3. Line 203-204: “In contrast, the cAMP level of FDHY23 at 46°C was obviously higher than its cAMP level at 30°C ($p = 0.213$).” The p value is more than 0.05, so the difference is insignificant. That is to say, there is no reliable difference between the two conditions. Also, in Lines 206-208, the authors state that “These results suggest that with the increase in temperature, FDHY23 enhances the production of cAMP during mid-log phase”, which is incorrect.

Response:

For strain FDHY23 in the mid-log phase, the average values of cAMP levels are 4.50 at 46°C and 3.83 at 30°C, with the p-value of 0.213 and Cohen’s d value of 1.67. Cohen’s d value is a measure of effect size, commonly used to assess the practical magnitude of the difference between two groups. A large effect size is typically considered to be Cohen’s d value above 0.8, indicating a substantial difference between the two groups. Therefore, although the p value is not significant, the large Cohen’s d value suggests the higher cAMP level in the mid-log phase of FDHY23 at 46°C compared to 30°C is meaningful, rather than simply due to chance. Based on these findings, we concluded that “These results suggest that with the increase in temperature, FDHY23 may enhance the production of cAMP during mid-log phase”. The corresponding revised sentences have been highlighted in blue in the manuscript. We apologize for not providing the Cohen’s d value earlier, which led to this confusion.

4. There are a few grammatical errors:

(1) Line 27: “CYR1 mutation induced reduction in cAMP levels” should be “CYR1 mutation-induced reduction in cAMP levels” .

(2) Line 55: “gene expressions” should be “gene expression” .

(3) Line 65: “researches” should be “research” .

(4) Line 175: “remarkable higher” should be “remarkably higher” .

Response:

Thanks a lot for the reminding, we have made these corrections in the revised manuscript.